

# Turbulence parameters measured by the Beijing Mesosphere–Stratosphere–Troposphere radar in the troposphere/lower stratosphere with three models: Comparison and analyses

Ze Chen[1, 2, 3], Yufang Tian [1, 2, 3, *], Yinan Wang[1, 2, 3], Yongheng Bi[1, 2, 3], Xue Wu[1, 2, 3], Juan Huo[1, 2, 3], Linjun Pan[1, 2, 3], Yong Wang [1, 2, 3], Daren Lü[1, 2, 3]

[1]Key Laboratory of Middle Atmosphere and Global Environment Observation (LAGEO), Institute of Atmospheric Physics, Chinese Academy of Sciences, Beijing 100029, China
[2]Xianghe Observatory of Whole Atmosphere, Institute of Atmospheric Physics, Chinese Academy of Sciences, Xianghe 065400, China
[3]University of Chinese Academy of Sciences, Beijing 100049, China

*Correspondence to*: Tian Yufang (tianyufang@mail.iap.ac.cn)

**Abstract.** Based on the quality-controlled observational spectral width data of the Beijing Mesosphere–Stratosphere–Troposphere (MST) radar in the altitudinal range of 3–19.8 km from 2012 to 2014, this paper analyzes the relationship between the proportion of negative turbulent kinetic energy (N-TKE) and the horizontal wind speed/horizontal wind vertical shear domain, and gives the distributional characteristics of atmospheric turbulence parameters obtained by using different calculation models. Three calculation models of the spectral width method were used in this study—namely, the H model (Hocking, 1985), N-2D model (Nastrom, 1997) and D-H model (Dehghan and Hocking, 2011). The results showed that the proportion of N-TKE in the H model increases with the horizontal wind speed and/or the vertical shear of horizontal wind speed, up to 80%. When the horizontal wind speed is greater than 40 m·s$^{-1}$, the proportion of N-TKE in the H model is greater than 60%, and thus the H model is not applicable. When the horizontal wind speed is greater than 20 m s$^{-1}$, the proportion of N-TKE in the N-2D model and D-H model increases with the horizontal wind speed, independent of the vertical shear of the horizontal wind speed, and the maximum values are 2% and 4%, respectively. However, it is still necessary to consider the applicability of the N-2D model and D-H model in some weather processes with strong winds. The distributional characteristics with height of the turbulent kinetic energy dissipation rate $\varepsilon$ and the vertical eddy diffusion coefficient Kz derived by the three models are consistent with previous studies. Still, there are differences in the values of turbulence parameters. Also, the range resolution of the radar has little effect on the differences in the range of turbulence parameters' values. The median values of $\varepsilon$ in the H model, N-2D model and D-H model are $10^{-3.2}$–$10^{-2.8}$ m$^2$ s$^{-3}$, $10^{-2.8}$–$10^{-2.4}$ m$^2$ s$^{-3}$ and $10^{-3.0}$–$10^{-2.5}$ m$^2$ s$^{-3}$, respectively. The median values of Kz in these three models are $10^{0.18}$–$10^{0.67}$ m$^2$ s$^{-1}$, $10^{0.57}$–$10^{0.90}$ m$^2$ s$^{-1}$ and $10^{0.44}$–$10^{0.74}$ m$^2$ s$^{-1}$.





## 1. Introduction


Small-scale turbulence plays a vital role in the vertical exchange of heat, momentum and mass in the atmosphere. Originally, observing turbulence in the free atmosphere was mainly carried out by sounding balloons and aircraft (e.g., Lilly et al., 1974). However, with the development of atmospheric radar, it has since become possible to quantitatively calculate turbulence parameters (e.g., the vertical eddy diffusion coefficient Kz and turbulence energy dissipation rate $\varepsilon$) in the free

atmosphere through remote sensing (Weinstock, 1981; Hocking, 1983).

Most research on turbulence parameters using atmospheric radar are based on the Kolmogorov hypothesis of isotropic turbulence at the inertial sub-region scale (Batchelor, 1953; Tatarski, 1961, 1971). To detect atmospheric turbulence intensity by atmospheric radar, the radar echo signal should come from turbulence scattering. In fact, at some heights, such as near the tropopause region, the scattering echo can be affected by specular reflection. However, the influence of specular reflection is

weaker for inclined beams than for vertical beam. Therefore, it is more appropriate to use the observational data of inclined beams for analysis. The Doppler spectrum width measured by radar contains atmospheric turbulence intensity information, and the turbulence is on a smaller scale than the radar sampling volume.

The Mesosphere–Stratosphere–Troposphere (MST) radar is a unique and essential means to detect turbulence characteristics in multiple layers of the atmosphere. As a kind of atmospheric radar, MST radar is based on the scattering effect

of atmospheric refraction irregularities on the electromagnetic waves emitted by the radar to carry out remote sensing detection of the atmosphere. Therefore, the radar echo contains atmospheric turbulence information (such as echo power and spectral width, etc.). Also, the scale of the detection target is in the inertial sub-region. For the current detection methods, MST radar is an indispensable instrument to detect the troposphere, stratosphere and mesosphere. The macroscopic characteristic parameters ($\varepsilon$, Kz) used to describe atmospheric turbulence are calculated using MST radar data with high spatial and temporal

resolution. At present, three methods are mainly used: the power method (Hocking, 1985), the Doppler spectral width method (Hocking, 1985; Nastrom, 1997; Dehghan and Hocking, 2011; Fukao et al., 2014), and the vertical velocity variance method (Satheesan and Murthy, 2002).

The basic idea of the power method is that the radar echo power can be used to estimate the structure constant of the atmospheric refractive index $C_n^2$ (Rao et al., 2001b), and the mathematical relationship between $C_n^2$ and $\varepsilon$ can be determined

by the outer scale of turbulence. Therefore, the turbulence parameters $\varepsilon$ and Kz can be calculated by the radar echo power. $\varepsilon$ has a mathematical relationship with the variance of vertical velocity ($\overline{\omega^2}$): $\varepsilon = \frac{6.1 F \overline{\omega^2} N}{2\pi} = 0.97 \overline{\omega^2} N$, where F is the fraction of the measured velocity variance (of wind velocity spectrum) that resides in the inertial subrange and the rest in the buoyancy subrange, and $N$ is the Brunt–Väisälä (B–V) frequency. Satheesan and Murthy (2002) have taken F = 1. The power method requires temperature, atmospheric pressure, and water vapor profile data, as well as the assumption that the radar absolute

calibration and radar detection volume are filled with turbulence. The vertical velocity law requires precise vertical velocity. For vertical beams, due to the interference of non-turbulent signals, the accuracy of vertical velocity needs to be improved. Delage et al. (1997) compared the statistical characteristics of $\varepsilon$ with the power method and the spectral width method,





separately. The results showed that the results of the two methods are in good agreement when the turbulent layer is thinner than 600 m.

For the spectral width method, the conditions of the above two methods are not necessary. Radar echo is the backscattering result of all scattering cells in the radar sampling space. For a given range library, due to coherent integration and incoherent integration of the radar, the random motion of the scattering cells is shown as the random distribution of its Doppler velocity near the mean wind speed. That is, the Doppler spectrum of the radar is broadened. The Doppler spectral width contains atmospheric turbulence information and can be used to calculate the macro parameters of turbulence.

The present study shows that the spectrum width $\sigma_o$ in the radar power spectrum has a turbulent contribution $\sigma_t$ and non-turbulent contribution $\sigma_u$, such as beam broadening $\sigma_b$ and shear broadening $\sigma_s$, under the condition of no interference signal:

$$\sigma_o^2 = \sigma_t^2 + \sigma_u^2 = \sigma_t^2 + \sigma_s^2 + \sigma_b^2 + \sigma^2, \tag{1}$$

where $\sigma^2$ refers to the influence of other factors, such as gravity waves, which will also cause the spectral width to increase in the total acquisition time of the radar. However, the contribution of $\sigma^2$ is relatively small in the region below 20 km, where

$\sigma_s^2 + \sigma_b^2$ can be combined into a term $\sigma_{s\&b}^2$, which represents beam and shear effects (Nastrom, 1997).

    In current studies, there are mainly three models used to calculate non-turbulent spectral width: Hocking (1983, 1985) proposed an empirical model (called the H model); Nastrom (1997) put forward a calculation model and revealed that their 2D model could meet the estimation requirements (called the N-2D model); and Dehghan and Hocking (2011) made a further derivation of the N-2D model and thus developed a new calculation model (called the D-H model). The three models are

described in detail in Section 2.3.

    Due to the differences in the calculation models of turbulence spectral width, the specific equations for calculating turbulence parameters using the spectral width method are different, but they have similar expressions. The relation between the turbulent energy dissipation rate $\varepsilon$ and $\sigma_t^2$ is as follows (Hocking, 1983; Weinstock, 1981):

$$\varepsilon = c_1 \sigma_t^2 N, \tag{2}$$

where $c_1$ is a constant and $N$ is the B–V frequency ($s^{-1}$). For the H model, $c_1$ varies in different studies, generally ranging from 0.45 to 0.5 (e.g., Hocking, 1999; Wilson, 2004). Hocking et al. (2016) suggested that $0.5 \pm 0.25$ was a reasonable range for $c_1$. For the H-model, this paper takes $c_1 = 0.45$, and Hocking (1999) obtains it from experience (Kohma et al., 2019). For the N-2D model, the turbulence in the inertial subregion is assumed to be isotropic. For a stably stratified atmosphere, $\sigma_t^2$ has the following relationship with $\varepsilon$ (Weinstock, 1981; Nastrom and Eaton, 1997): $\varepsilon = A^{-\frac{3}{2}} N \sigma_t^2$, where $A$ is the Kolmogorov constant,

taking A = 1.6, $c_1 \approx 0.49$. $N^2 = g \frac{dln(\theta)}{dz}$, and the potential temperature $\theta$ can be calculated by the equation $\theta = T \left( \frac{1000}{P} \right)^{0.286}$, where $T$ is the temperature (K) and $P$ is atmospheric pressure (hPa). $\theta$ can be calculated from the radiosonde data.

    Kz is closely related to $\varepsilon$ (Fukao et al., 1994; Nastrom and Eaton, 1997; Rao et al., 2001a). The equation is as follows:

$$K_z = c_2 \varepsilon N^{-2} = c_1 c_2 N^{-1} \sigma_t^2, \tag{3}$$





where $\varepsilon$ is the dissipation rate of turbulent energy, $N$ is the B–V frequency, and $c_2$ is a constant. In this paper, $c_2 = 0.3$ (Fukao
et al., 1994).

When the spectral width method is used to calculate the turbulence parameters, there is a negative value of $\sigma_t^2$ in the results of the H, N-2D and D-H models, resulting in negative values of the turbulence parameters $\varepsilon$ and Kz—that is, negative turbulent kinetic energy (N-TKE). Dehghan and Hocking (2011) believed that the factors that cause the negative value of the turbulent spectrum width mainly include the non-isotropy of the scatterer (relatively small contribution), the influence of the
uncertainty of the calculation of the observed spectrum width, and the spectrum width broadening term (Eq. 1). The $\sigma_o^2$ is related to the calculation method of each moment of the power spectrum and the resolution of the power spectrum (depends on the data length, s), while $\sigma_{s\&b}^2$ depends on the uncertainty of the calculation of horizontal wind speed. When the $\sigma_o^2$ value is low and the $\sigma_{s\&b}^2$ value is high, $\sigma_t^2$ will be low, sometimes even negative; and when $\sigma_o^2$ is high and $\sigma_{s\&b}^2$ is low, $\sigma_t^2$ will be high. Kohma et al. (2019) pointed out that the median of $\varepsilon$ differs slightly ($< 3\%$) between including and excluding negative
numbers.

Since the influence of non-isotropy is relatively small, for a radar (assuming constant radar parameters), Eq. (1) can be simplified as $\sigma_o^2 = \sigma_t^2 + \sigma_{s\&b}^2$ in the tropospheric and lower stratospheric range. The main factor causing $\sigma_t^2 < 0$ is the calculation accuracy of $\sigma_{s\&b}^2$. If the radar parameter is constant, the factors affecting the calculation accuracy of $\sigma_{s\&b}^2$ are not only the accuracy of the calculation of the horizontal wind field (the horizontal wind speed and the vertical shear of horizontal
wind), but also the applicability of the calculation model itself may be different under different horizontal wind field conditions. For example, Dehghan and Hocking (2011) believed that in some strong wind shear conditions, a more universal model than the D-H model is needed. When the probability of N-TKE is high, the applicability of the model is the main factor affecting the calculation accuracy of $\sigma_{s\&b}^2$. Moreover, when the amount of data involved is statistically too small, the credibility of the final turbulence parameter structure will be reduced. Therefore, before analyzing the turbulence parameters, the applicability
of the non-turbulent spectral width calculation model in different horizontal wind fields should be analyzed.

Based on three years of observational data from the Beijing MST radar (2012, 2013 and 2014), this paper uses three models to calculate the non-turbulent spectrum width and analyzes the distributional characteristics of the N-TKE ratio under different horizontal wind speeds and horizontal wind vertical shear conditions. It can also be understood as the frequency distribution characteristics of horizontal wind speed and vertical shear of horizontal wind speed when N-TKE appears.
Furthermore, the vertical distribution characteristics of the turbulence parameters are analyzed, and the applicability of the three models is given. By studying the applicability of the calculation models in the different wind field conditions, the appropriate model can be selected to calculate the non-turbulent spectrum width to improve the reliability of the calculation results of turbulence parameters.

This remainder of the paper is structured as follows: Section 2 describes the data and methods, in which the three models
used to calculate non-turbulent broadening are outlined. In section 3, the relationship between the occurrence probability of N-TKE and horizontal wind speed/the vertical shear of horizontal wind speed along with the analysis results of the





distributional characteristics of turbulence parameters are given. Sections 4 and 5 are the discussion and conclusion, respectively.



## 2 Data and methods

### 2.1 Beijing MST radar observations

The data used in this paper are the observational data of the Beijing MST radar, which is located at the Xianghe Observatory of the Whole Atmosphere, Institute of Atmospheric Physics, Chinese Academy of Sciences (39.78°N, 116.95°E). The Beijing MST radar is a five-beam (east–west, north–south and vertical) clear air turbulence (CAT) detection pulse Doppler radar, which was built and put into service in 2011 and has accumulated a long period of data. According to analyses of the reliability and accuracy of the Beijing MST radar data (Tian and Lü, 2016, 2017), it has good detection capability in the troposphere, lower stratosphere, and mesosphere to lower thermosphere. Tian and Lü (2017) described the Beijing MST radar in more detail. The parameters of the Beijing MST radar are shown in Table 1.

**Table 1. Parameters of the Beijing MST radar.**

| Parameter | Value | |
|---|---|---|
| Location | Xianghe Station, China (39°45′14.40″N, 116°59′24.00″E) | |
| Operating frequency | 50±1 MHz | |
| Number of beams | 5 (E, W, S, N, H) | |
| Peak power output | 172.8 kW | |
| Half-power full-beam width | 3° | |
| | Low-mode | Mid-mode |
| Zenith angle of oblique | 15° | 15° |
| Coherent integration | 128 | 64 |
| Incoherent integration | 10 | 10 |
| Number of FFT | 256 | 256 |
| Pulse length | 1 μs | 32 μs |
| Interpulse period | 160 μs | 320 μs |
| Range resolution | 150 m | 600 m |

This paper uses data from four tilted beams (east–west, north–south) with a zenith angle of 15°. The radial range resolutions of mid-mode and low-mode observations are 600 m and 150 m, respectively. The advantage of using vertical beam detection results to calculate turbulence parameters is that the influence of wind shear does not need to be considered (Kantha et al., 2017). However, the vertical beam is more susceptible to specular reflection, such as a statically stable tropopause region. At the same time, because the radial velocity of the vertical beam is small, it is more affected by ground clutter near zero





frequency, which reduces the accuracy of vertical beam spectrum observations. Compared with the vertical beam, the oblique beam is less likely to be affected by specular reflection than by isotropic scattering due to isotropic turbulence (Fukao et al., 1994; Tsuda et al., 1986). Therefore, based on the above considerations, this paper uses the spectral width data obtained from the four tilted beams to calculate the turbulence parameters. In this paper, the improved power spectral density processing

algorithm of Chen et al. (2020) is applied to suppress non-atmospheric signals and obtain reliable spectral width data effectively.

**2.2 Radiosonde data**

For the spectral width method, $N^2$ profiles need to be provided in other ways when turbulence parameters are calculated by the turbulent spectral width. In this paper, the temperature profile data of the Beijing conventional radiosonde (54511, 39.8°N, 116.4°E) are used to calculate $N^2$. The straight-line distance between the MST radar and the radiosonde launch

site is about 40 km. Conventional radiosonde probes are operated twice a day (11:15 and 23:15 UTC) and recorded every 1–2 s, with a vertical resolution of about 10 m. In this paper, the observational data of the mid-mode (11:10, 11:40, 23:10 and 23:40 UTC) and low-mode (11:05, 11:35, 23:05 and 23:35 UTC) of the Beijing MST radar from 2012 to 2014, corresponding to the radiosonde, are selected to calculate the turbulence parameters. The number of radiosonde profiles involved in the calculation of both the mid and low modes is 3532. The radiosonde data are interpolated with a resolution of 600 m in the

radar mid-mode to facilitate the calculation. In low observation mode, the radiosonde data are interpolated with a resolution of 150 m.

**2.3 Methods used to estimate turbulence parameters**

In the troposphere–lower stratosphere region, time broadening (also called the gravity wave term) has a relatively small effect on the observed spectrum width (Nastrom, 1997). The broadening of the spectrum caused by turbulence mainly considers

shear and beam broadening: $\sigma_t^2 = \sigma_o^2 - \sigma_s^2 - \sigma_b^2$. After calculating the radar observation spectrum width, we then estimate $\sigma_s^2$ and $\sigma_b^2$ to obtain $\sigma_t^2$. The atmospheric turbulence parameters ($\varepsilon$, Kz) can be estimated by $\sigma_t^2$ according to Eqs. (2) and (3). Based on this, there are currently several calculation models for calculating $\sigma_t^2$ by the spectral width method, and they have similar expressions.

Before introducing the three calculation models, due to the differences in expression between the models, it is necessary

to understand the relationship between the power spectrum half-power half-width ($\sigma_{\frac{1}{2}}$) and the Doppler spectrum width ($\sigma$), $\sigma = \frac{\sigma_{\frac{1}{2}}}{\sqrt{2\ln 2}}$). The units of $\sigma$ and $\sigma_{\frac{1}{2}}$ can be Hz or $m \cdot s^{-1}$. The relationship between the Doppler velocity $v$ and the Doppler frequency shift $f$ is as follows: $v = f \cdot \lambda / 2$, where $\lambda$ is the wavelength of the electromagnetic wave emitted by the radar. The Doppler velocity spectrum width $\sigma_v$ (or the radial velocity standard deviation) and the Doppler frequency spectrum width $\sigma_f$ have the following relationship: $\sigma_v = \frac{\lambda}{2}\sigma_f$. Similarly, $\sigma_{v\frac{1}{2}} = \frac{\lambda}{2}\sigma_{f\frac{1}{2}}$, where $\sigma_{v\frac{1}{2}}$ and $\sigma_{f\frac{1}{2}}$ are the Doppler velocity and half-power

and half-width (Hz), respectively.





### 2.3.1 H-model

According to Hocking (1985), the beam broadening can be estimated using the following equation:

$$\sigma_{vb} = \sigma_{f\frac{1}{2}b} \cdot \frac{\lambda}{2} / (\sqrt{2}ln2)$$

$$= (1.0) * \frac{2}{\lambda} \cdot \theta_{\frac{1}{2}}V \cdot \frac{\frac{\lambda}{2}}{\sqrt{2\ln 2}}$$

$$= (1.0) * \frac{\theta_{\frac{1}{2}}V}{\sqrt{2\ln 2}}, \tag{4}$$


where $\sigma_{vb}$ is the Doppler velocity spectrum width caused by the beam ($m \cdot s^{-1}$), $\sigma_{f\frac{1}{2}b}$ is the half-power half-width (HZ) of the

Doppler frequency caused by the beam, $\sigma_{f\frac{1}{2}b} = (1.0) \times \frac{2}{\lambda}\theta_{\frac{1}{2}}V$, where $\lambda$ is the wavelength of the electromagnetic wave emitted

by the radar (the $\lambda$ of the Beijing MST radar is 6 m), and $\theta_{\frac{1}{2}}$ is the effective beam width of the radar. $\theta_{\frac{1}{2}}$ is the two-way (transmit

and receive) half-power half-width in the polar coordinate system (Hocking et al., 2016, Eq. 7.34). The $\theta_{\frac{1}{2}}$ of the Beijing MST

radar is $\frac{1.5}{180} \times \pi$ (radians), and $V$ is the average horizontal wind speed (m s$^{-1}$) calculated by tilting the beam.

Wind shear widening can be calculated with the following equation (Hocking 1985; Fukao et al., 2014):

$$\sigma_{vs} = \frac{\sigma_{v\frac{1}{2}s}}{\sqrt{2ln2}} = \frac{1}{2} \cdot \frac{\left|\frac{\partial u}{\partial z}\right| \sin(\chi)\Delta r}{\sqrt{2ln2}}, \tag{5}$$

where $\sigma_{vs}$ is the widening of the Doppler velocity spectrum caused by the vertical shear of the horizontal wind and $\sigma_{v\frac{1}{2}s}$ is the

half-power half-width (m s$^{-1}$) caused by the horizontal wind shear. $\sigma_{v\frac{1}{2}s} = \frac{1}{2} \cdot \left|\frac{\partial u}{\partial z}\right| \sin(\chi) \Delta r$, where $\left|\frac{\partial u}{\partial z}\right|$ is the horizontal wind

and vertical shear, $\chi$ is the zenith angle of the beam, and $\Delta r$ is radial resolution of the radar.

In this paper, Eq. (4) and Eq. (5) are referred to as the H model for short. For the vertical beam ($\chi = 0°$), the value of the

broadening term caused by wind shear is zero, so Eq. (4) can be used to calculate the $\sigma_{s\&b}^2$ of the vertical beam. The effect of

beam broadening can be processed before obtaining the power spectrum. For example, the PANSY radar uses irregular

antennas, and deconvolution is performed before the power spectrum is obtained. Therefore, when using radar data to calculate

turbulence parameters, there is no need to consider beam broadening (Fukao et al., 2014; Kohma et al., 2019).

Incorporating Eq. (4) and Eq. (5) into the equation $\sigma_t^2 = \sigma_o^2 - \sigma_s^2 - \sigma_b^2$ allows $\sigma_t^2$ to be calculated. Since the turbulence

in the inertial subregion satisfies the hypothesis of specific isotropy, the variance $\overline{v^2}$ (or turbulent energy) of the scatterer's

wind speed fluctuation and the turbulence spectrum width $\sigma_{tur}^2$ have the following relationship:

$$\overline{v^2} = \sigma_t^2 = \sigma_{vo}{}^2 - \left((1.0) * \theta_{1/2}V/\sqrt{2ln2}\right)^2 - \left(\frac{1}{2} \cdot \left|\frac{\partial u}{\partial z}\right| \frac{\sin(\chi)\Delta r}{\sqrt{2ln2}}\right)^2, \tag{6}$$

where $\sigma_{vo}$ is the observed Doppler velocity spectrum width (m s$^{-1}$) and $\sigma_{vo}$ can be calculated by Gaussian fitting.





### 2.3.2 N-2D model

Nastrom (1997) and others believe that their 2-dimensional model can describe well the broadening of the spectral width caused by the beam and horizontal wind shear (referred to as the N-2D model). The N-2D model considers the effects of beam and shear at the same time. That is, $\sigma_s^2$ and $\sigma_b^2$ in Eq. (1) are combined into a term $\sigma_{s\&b}^2$. The equation for broadening the spectral width is as follows:

$$\sigma_{s\&b}^2 = \frac{\theta^2}{3} v^2 cos^2\chi - \frac{2\theta^2}{3} sin^2\chi \left(v\frac{\partial v}{\partial z} rcos\chi\right)/24 \left(3 + cos4\chi - 4cos2\chi\right) \left(\frac{\partial v}{\partial z}\right)^2 r^2$$
$$+ \left(\frac{\theta^2}{3} cos4\chi + sin^2\chi cos^2\chi\right)\left(\frac{\partial v}{\partial z}\right)^2 \frac{\Delta r^2}{12}, \tag{7}$$

where $\theta$ is the half power and half width (radians) of the radar beam, $v$ is the horizontal wind speed, $\frac{\partial v}{\partial z}$ is the vertical shear of the horizontal wind speed, $\chi$ is the zenith angle, $r$ is the distance, and $\Delta r$ is the radar resolution.

### 2.3.3 D-H model

In the study of non-turbulent flow broadening the spectrum, Dehghan and Hocking (2011) gave a new calculation model (referred to as the D-H model) based on the N-2D model. Compared with the N-2D model, the D-H model has one more item, making the result more accurate. The specific equation is as follows:

$$\sigma_{s\&b}^2 = \frac{\theta^2}{k} v^2 cos^2\chi - a_0 \frac{\theta}{k} sin\chi \left(v\frac{\partial v}{\partial z}\zeta\right) + b_0 \frac{2sin^2\chi}{8k} \left(\frac{\partial v}{\partial z}\zeta\right)^2$$
$$+ c_0(cos^2\chi sin^2\chi)|v\xi| + d_0 \left(cos^2\chi sin^2\chi\right)\xi^2, \tag{8}$$

where $k = 4ln2$, $\zeta = 2r\theta sin\chi$, $\xi = \frac{\partial v}{\partial z}\frac{\Delta r}{\sqrt{12}}$, $a_0 = 0.945$, $b_0 = 1.500$, $c_0 = 0.030$, and $d_0 = 0.825$.

If the non-isotropy of the scatterer is not considered (the contribution is relatively small), the accuracy of the calculation of $\sigma_o^2$ and $\sigma_{s\&b}^2$ will directly cause $\sigma_t^2$ to be too small or too large. From the equations of the three calculation models [Eqs. (6), (7) and (8)], if the radar parameters are constant, after using Gaussian fitting to calculate the moments of the power spectrum, and assuming that the calculated observational spectrum width has a small contribution to $\sigma_t^2$ less than zero, the accuracy of $\sigma_{s\&b}^2$ is the main factor causing N-TKE. In certain horizontal wind field conditions (horizontal wind speed $u$ and horizontal wind vertical shear $\frac{du}{dz}$), when the probability of occurrence of N-TKE is high, the applicability of the calculation model is the main factor affecting the accuracy of $\sigma_{s\&b}^2$.

## 3. Results

### 3.1 Relationships between N-TKE rates and both the horizontal wind and vertical shear of horizontal wind

Using the Beijing MST radar to detect data within the range of 3–19.8 km in the middle observation mode from 2012 to 2014, we counted the total number of effective values of the observational spectrum width and the total number of $\sigma_t^2 < 0$





values calculated by the three models used, as shown in Table 2. The results show that for the D-H model, N-2D model and H model, the rate of N-TKE ($\sigma_t^2 < 0$) is in the range of 0.8%–1%, 0.4%–0.5% and 30%–40%, respectively. The probability that

the turbulence spectrum width is less than 0 calculated by the H model is significantly higher than that of the other two models.

We further analyzed the two-dimensional frequency distribution characteristics of horizontal wind speed and the vertical shear of horizontal wind speed in the range of 3–19.8 km above the radar station when the spectrum width value detected by the radar was valid, as shown in Fig. 1. The horizontal wind speed in Beijing is distributed between 0 m s$^{-1}$ and 60 m s$^{-1}$, mainly concentrated in the range of 0–40 m s$^{-1}$. The vertical shear of the horizontal wind speed ranges from −0.012 to 0.012

s$^{-1}$, concentrated primarily in the range of −0.004 to 0.006 s$^{-1}$.

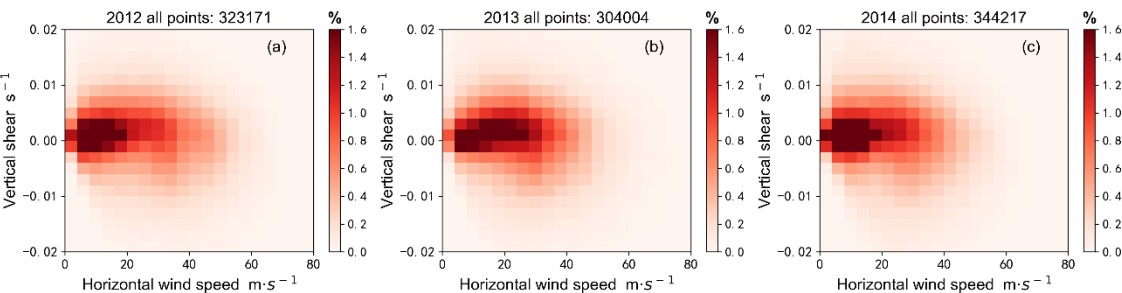

**Figure 1. Two-dimensional frequency distribution characteristics of horizontal wind speed and vertical shear of horizontal wind speed within the height range of 3–19.8km above the Beijing MST radar station from 2012 to 2014.**

**3.1.1 Probability distribution characteristics of horizontal wind versus the vertical shear of horizontal wind observed**
**by the Beijing MST radar**

We further analyzed the distributional characteristics of the horizontal wind speed and vertical shear of horizontal wind speed in the case of the N-TKE calculated by the three models, as shown in Fig. $2(a_1, a_2, a_3)$ and $(b_1, b_2, b_3)$. Meanwhile, Fig. $2(c_1, c_2, c_3)$ shows the distributional characteristics of the three different models, $R^- \left( u, \frac{du}{dz} \right)$, in the horizontal wind speed $u$ and horizontal wind vertical shear $\frac{du}{dz}$ domain. That is, the two-dimensional frequency distribution characteristics of $u$ and $\frac{du}{dz}$

in the Beijing area when $\sigma_t^2 < 0$. $R^- = \frac{n_{ij}}{N^-}$, where $n_{ij}$ is the frequency with negative $\sigma_t^2$ in a certain grid cell $(u_i \to u_{i+1}, \frac{du}{dz_j} \to \frac{du}{dz_{j+1}})$ and $N^-$ is the total frequency of negative $\sigma_t^2$, as shown in Table 2. Three years of data from the Beijing MST radar from 2012 to 2014 are used.

**Table 2. Total frequency of $\sigma_t^2 < 0$ in the range of 3–19.8 km.**

| Time | All | D-H, $\sigma_t^2 < 0$ | N-2D, $\sigma_t^2 < 0$ | H , $\sigma_t^2 < 0$ |
|------|------|------|------|------|
| 2012 | 323171 | 2711 (0.84 %) | 1356 (0.42%) | 122589 (37.93%) |
| 2013 | 304004 | 2992 (0.98%) | 1532 (0.50%) | 112290 (36.94%) |





| 2014 | 344217 | 3013 (0.88%) | 1712 (0.50%) | 105276 (30.58%) |



**Figure 2.** Frequency distribution of (a$_1$–a$_3$) horizontal wind speed and (b$_1$–b$_3$) the vertical shear of horizontal wind speed, along with (c$_1$–c$_3$) the two-dimensional frequency distribution characteristics of horizontal wind speed and the vertical shear of horizontal wind speed for **H model** (a$_1$, b$_1$, c$_1$), **N-2D model** (a$_2$, b$_2$, c$_2$) and **D-H model** (a$_3$, b$_3$, c$_3$) when the turbulent kinetic energy is negativie.

As shown in Fig. 2(a$_1$, b$_1$), the medians of $u$ and $\frac{du}{dz}$ of the H model are about 33.9 m s$^{-1}$ and 0 s$^{-1}$, respectively. The $u$ and

$\frac{du}{dz}$ are respectively distributed within 0 to 80 m s$^{-1}$ and −0.04 to 0.04 s$^{-1}$, where the frequency distribution of $\frac{du}{dz}$ manifests as





a rightward heavy-tail feature. As shown in Fig. 2($a_2$, $b_2$), the medians of $u$ and $\frac{du}{dz}$ of the N-2D model are about 41.5 m s⁻¹ and

0 s⁻¹, respectively. The $u$ and $\frac{du}{dz}$ are respectively distributed within 10 to 80 m s⁻¹ and −0.04 to 0.02 s⁻¹, where the frequency

distribution of $u$ has a heavy-tailed distribution that is obviously to the left, and the frequency distribution of $\frac{du}{dz}$ appears as a

rightward heavy-tailed distribution. As shown in Fig. 2($a_3$, $b_3$), the medians of $u$ and $\frac{du}{dz}$ of the D-H model are about 42.2 m

s⁻¹ and 0 s⁻¹, respectively. The frequency distribution characteristics of $u$ and $\frac{du}{dz}$ are relatively consistent with those of the N-

2D model.

The $N^-$ value of the H model (total number of $\sigma_t^2 < 0$ values) is greater than that of the N-2D and D-H models, but the

$R^-$ values of the three models are all within the range of 10 to 70 m s⁻¹ and −0.02 to 0.02 s⁻¹, as shown in Fig. 2($c_1$, $c_2$, $c_3$).

Compared with the H model, the distributions of $\frac{du}{dz}$ of the N-2D model and the D-H model are more concentrated, and $u$ is

positively skewed. We also analyzed the wind field distribution characteristics of different years (2012, 2013 and 2014) when

$\sigma_{tur}^2 < 0$, and the results are similar to those in Fig. 2 (figures not shown).

### 3.1.2 Distributional characteristics of negative $\sigma_{tur}^2$ for the three methods

As shown in Fig. 2($c_1$, $c_2$, $c_3$), when the three models are used to calculate the turbulence spectrum width in Beijing, the

values of $R^-$ are significantly different in different ranges of $u$ and $\frac{du}{dz}$. That is, the probability of N-TKE has a different

dependence on horizontal wind speed and the vertical shear of horizontal wind speed.

Due to the specific locality of the wind field distribution characteristics, the total samples of each grid cell ($u_i \rightarrow u_{i+1}$,

$\frac{du}{dz}_j \rightarrow \frac{du}{dz}_{j+1}$) in Fig. 2($c_1$, $c_2$, $c_3$) are different. To analyze the universal relationship between the probability of N-TKE and

both the horizontal wind speed and vertical shear in the three models, it is necessary to consider the difference in the amount

of total samples. Therefore, we further statistically analyzed the probability of occurrence of N-TKE in each region of

horizontal wind speed and vertical shear of horizontal wind speed ($R_a^-$) calculated by the three models in each year of 2012–

2014, as shown in Fig. 3. The definition of $R_a^-$ is $R_a^- = \frac{n_{ij}}{Na_{ij}}$, where $n_{ij}$ is the frequency of $\sigma_t^2 < 0$ and $Na_{ij}$ is the total

frequency for which $\sigma_t^2$ is a valid value in the grid cell ($u_i \rightarrow u_{i+1}, \frac{du}{dz}_j \rightarrow \frac{du}{dz}_{j+1}$).





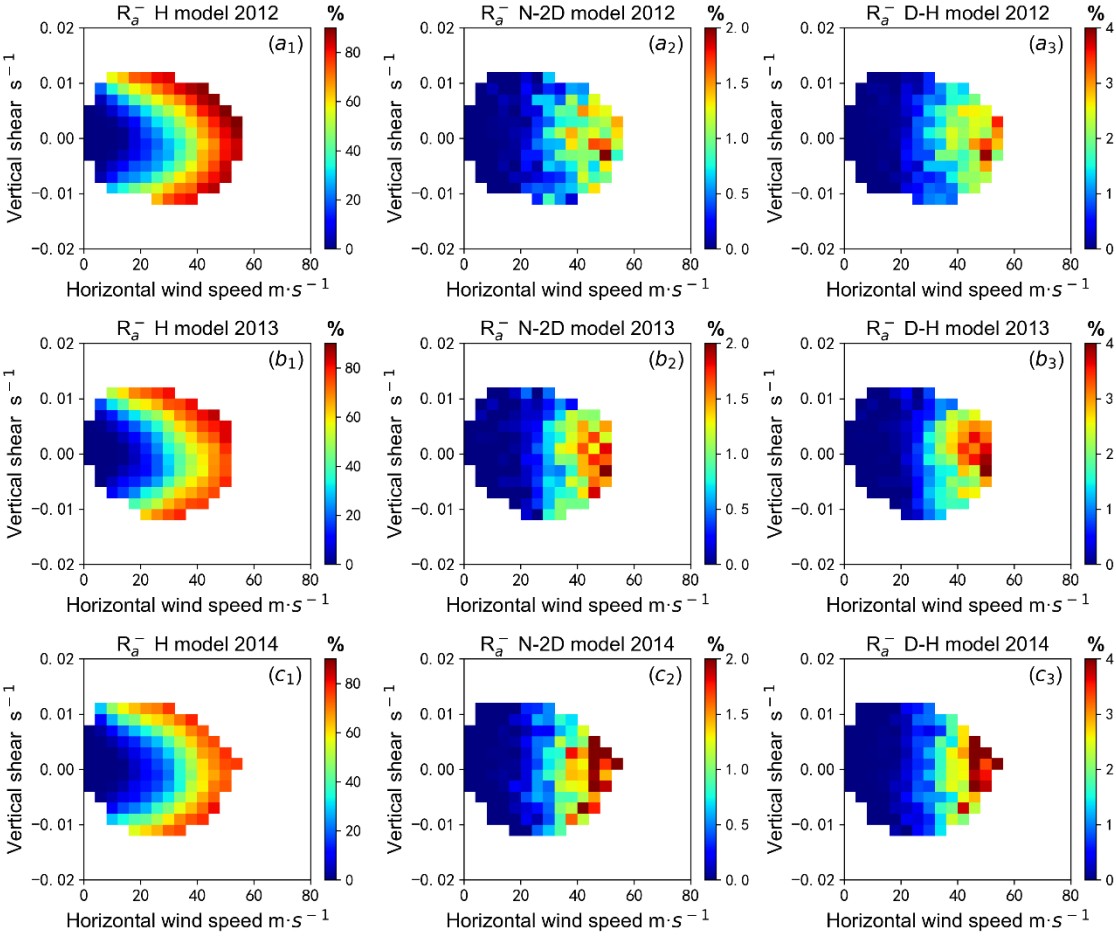

**Figure 3. Distribution of $R_a^-$ for the ($a_1$) H model, ($a_2$) N-2D model, and ($a_3$) D-H model in 2012. Panels ($b_1$)–($b_3$) and ($c_1$)–($c_3$) are the same as ($a_1$)–($a_3$) but for the results of the three models in 2013 and 2014, respectively.**

Based on mid-mode data of the Beijing MST radar, the distributional characteristics of the $R_a^-$ calculated by the three methods are shown in Fig. 3. The results show that the effective data rate of each area is greater than 0.2%. It can be seen that $\frac{du}{dz}$ is between −0.012 and 0.012 s⁻¹, and $u$ is between 0 and 60 m s⁻¹. Regardless of which model is used, the distributional characteristics of $R_a^-$ with ($u, \frac{du}{dz}$) in each year of 2012–2014 are consistent. The $R_a^-$ of the H model can reach 80%, and the probability of occurrence of N-TKE is significantly higher than that of the other two models. Furthermore, the $R_a^-$ of the N-2D model and the D-H model ranges from 0% to 2% and 0% to 4%, respectively.

For the N-2D model [Fig. 3($a_2, b_2, c_2$)] and D-H model [Fig. 3($a_3, b_3, c_3$)], $R_a^-$ is more sensitive to the horizontal wind speed, but does not change significantly with the vertical shear of the horizontal wind speed. When the horizontal wind speed is greater than 20 m s⁻¹, $R_a^-$ increases with the horizontal wind speed, the $R_a^-$ of the N-2D model is between 0.5% and 2%, and the $R_a^-$ of the D-H model is between 1% and 4%. When the horizontal wind speed is less than 20 m s⁻¹, $R_a^-$ does not change





significantly with the horizontal wind speed, the $R_a^-$ of the N-2D model is less than 0.5%, and the $R_a^-$ of the D-H model is less than 1%.

For the H model [Fig. 3($a_1$, $b_1$, $c_1$)], $R_a^-$ is sensitive to the magnitude of the horizontal wind speed and the vertical shear

of the horizontal wind speed. Also, $R_a^-$ increases with the horizontal wind speed and the absolute value of the vertical shear of the horizontal wind speed. When the vertical shear of the horizontal wind speed is between −0.005 and 0.005 $s^{-1}$, and the horizontal wind speed is less than 20 m $s^{-1}$, the $R_a^-$ has a relatively small value (< 20%). When the horizontal wind speed is greater than 40 m $s^{-1}$, the N-TKE of the H model accounts for more than 60%, and the H model is not applicable at this time.

### 3.2 Distributional characteristics of negative $\sigma_{tur}^2$ as a function of height for the three methods in Beijing

According to the above analysis, the three models for calculating the turbulence spectrum width have obvious differences in the dependence of the horizontal wind speed and the horizontal wind vertical shear. The radar site is located in the mid-latitude westerly zone in the northern hemisphere, and the horizontal wind field at each height has obvious seasonal changes. Therefore, we further analyzed the variational characteristics of the proportion of N-TKE with height in different seasons obtained by the three models, and provide a reference for better selection of applicable models. Based on the three years of

observational data from the Beijing MST radar, the annual average proportion of N-TKE and the average profile in February (winter) and July (summer) were obtained, as shown in Fig. 4.

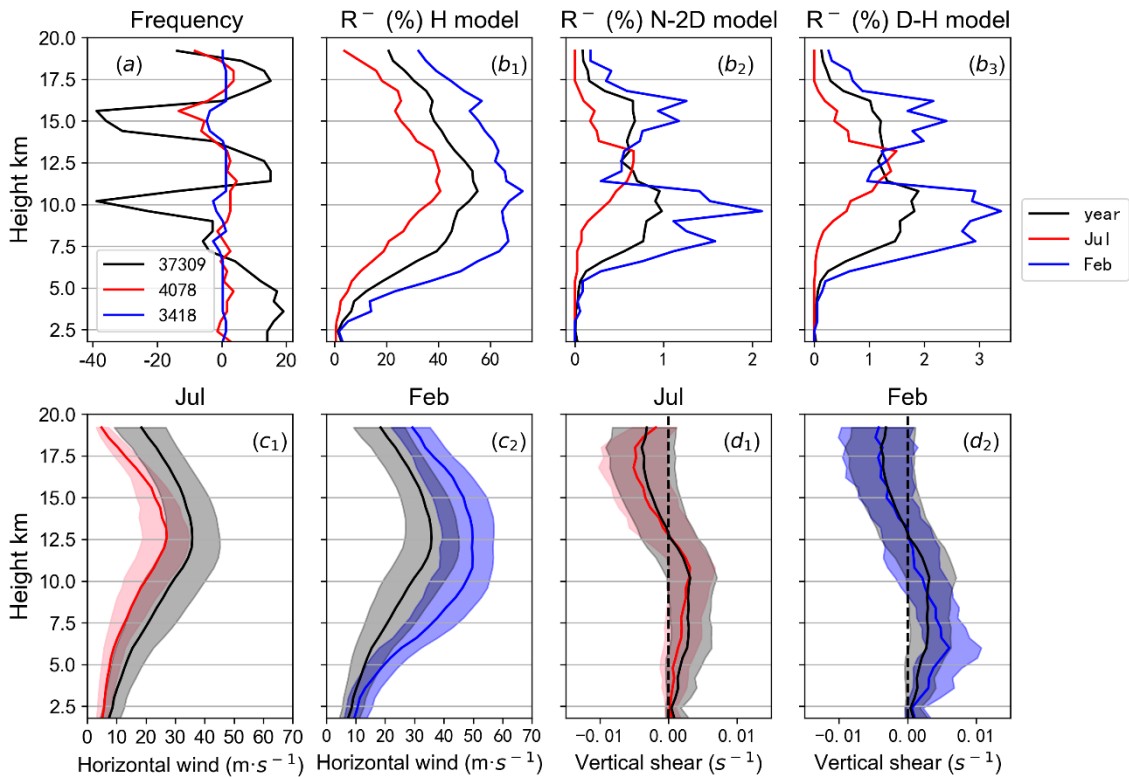



**Figure 4. (a) Deviation profile of the data volume involved in the statistics and the mean value of the profile. The annual mean value is 37,309, the mean value in July is 4078, and the mean value in February is 3418. ($b_1$–$b_3$) Probability of N-TKE in each gate for the H model, N-2D model and D-H model, respectively. Panels ($c_1$, $c_2$) and ($d_1$, $d_2$) are the median, upper and lower quartile profiles of horizontal wind speed and the vertical shear of horizontal wind speed, respectively. Black/red/blue represents the characteristics of the year/July/February, respectively. Three years of radar observational data from 2012 to 2014 were used in the statistics.**

As shown in Fig. 4a, within the range of 3–19.8 km, the average number of effective detections at all altitudes for the three years from 2012 to 2014 is 37,309, and the deviation of each altitude from the average is between –40 and 20. The average number of effective detections at all altitudes in July and February are 4078 and 3418, respectively, and the deviation of each altitude from the average is about –10 to 10. The difference in the number of effective detections at each height is very small, and the difference in the total sample size involved in the statistics does not affect the statistical results.

The annual average profile of the proportion of N-TKE calculated by the three methods is shown in Fig. 4($b_1$, $b_2$, $b_3$) (solid black line). The proportion of N-TKE first increases and then decreases with altitude. All three models have a maximum value area at 10–11 km. In this altitudinal range, the horizontal wind speed is large, and there is strong vertical shear [Fig. 4($c_1$, $c_2$, $d_1$, $d_2$)]. The maximum value of the ratio of N-TKE from the H model is 50%–60%; the maximum value of the N-2D model is about 1%; and the maximum value of the D-H model is about 2%. The ratio of N-TKE between the N-2D model and the D-H model has a second maximum at about 15 km.

For the N-2D model and D-H model, the annual mean and monthly mean (February and July) profiles of the rate of N-TKE are close to 0% in the region below 6 km, and the horizontal wind speed in this height range is less than 20 m s$^{-1}$. The result in Section 3.1 showed that when the horizontal wind speed is less than 20 m s$^{-1}$, the proportion of N-TKE is extremely low and does not change with the horizontal wind speed. In summer, the horizontal wind speed is less than 20 m s$^{-1}$ below 7.5 km, so the proportion of N-TKE in this range is close to 0%. In the range of 6–11 km and 13–16 km, compared with the annual average profile, the average profile of the proportion of N-TKE in winter increases from 1% to 2% and 1.5% to 3%, respectively. The average profile of the proportion of N-TKE of the turbulence in summer is reduced to less than 1%. This is mainly related to the fact that the horizontal wind speed is greater than 20 m s$^{-1}$ in the two altitudinal ranges, and the horizontal wind speed is larger in winter and smaller in summer in Beijing [Fig. 4($c_1$, $c_2$)]. Furthermore, the results in section 3.1 also showed that the proportion of N-TKE increases with the horizontal wind speed. Plus, also of note is that there is no significant difference between the annual profile and the monthly profile in the height range of 11–13 km, which is the maximum value of horizontal wind speed (the vertical shear of horizontal wind speed is close to 0 s$^{-1}$). When the horizontal wind speed and vertical shear are close to 0 s$^{-1}$, the first term on the right-hand side of Eq. (7) and Eq. (8) is the main contributing term, and the other terms on the right-hand side are close to zero.

For the H model, compared with the annual distribution, the proportion of N-TKE in winter increases over the entire height range, and the proportion of N-TKE decreases in summer. This is mainly related to the fact that the horizontal wind speed at all heights in winter is higher than that in summer, and the vertical shear profile of the horizontal wind speed in each season is the same at all altitudes. Section 3.1 showed that the proportion of N-TKE in the H model increases with the horizontal wind speed and the absolute value of the vertical shear of horizontal wind speed.


### 3.3 Annual mean profile of turbulence parameters estimated using the three methods

The proportion of N-TKE can be a reference for the selection of the turbulence spectrum width calculation model to some
extent. However, whether there are differences in the distributional characteristics of turbulence parameters calculated by the
three models of the spectral width method requires further analysis. From 2012 to 2014 in Beijing, the distributions at each
height of the observed spectral width, B-V frequency, turbulence dissipation rate obtained by three calculation models, vertical
turbulence diffusion coefficient, beam-shear broadening, and the distribution of spectral width caused by turbulence at each
height are shown in Fig. 5. The turbulence spectrum width contains negative values, as do $\varepsilon$ and Kz. The difference between
including negative values and excluding them is very small. The ratios of the median mean $\varepsilon$ calculated by the H model, N-2D
model and D-H model (including/excluding negative values) are $10^{-2.8}/10^{-2.7}$, $10^{-2.5}/10^{-2.5}$ and $10^{-2.6}/10^{-2.5}$, respectively. The
ratio of Kz is $10^{0.34}/10^{0.48}$, $10^{0.66}/10^{0.66}$ and $10^{0.53}/10^{0.56}$, respectively.

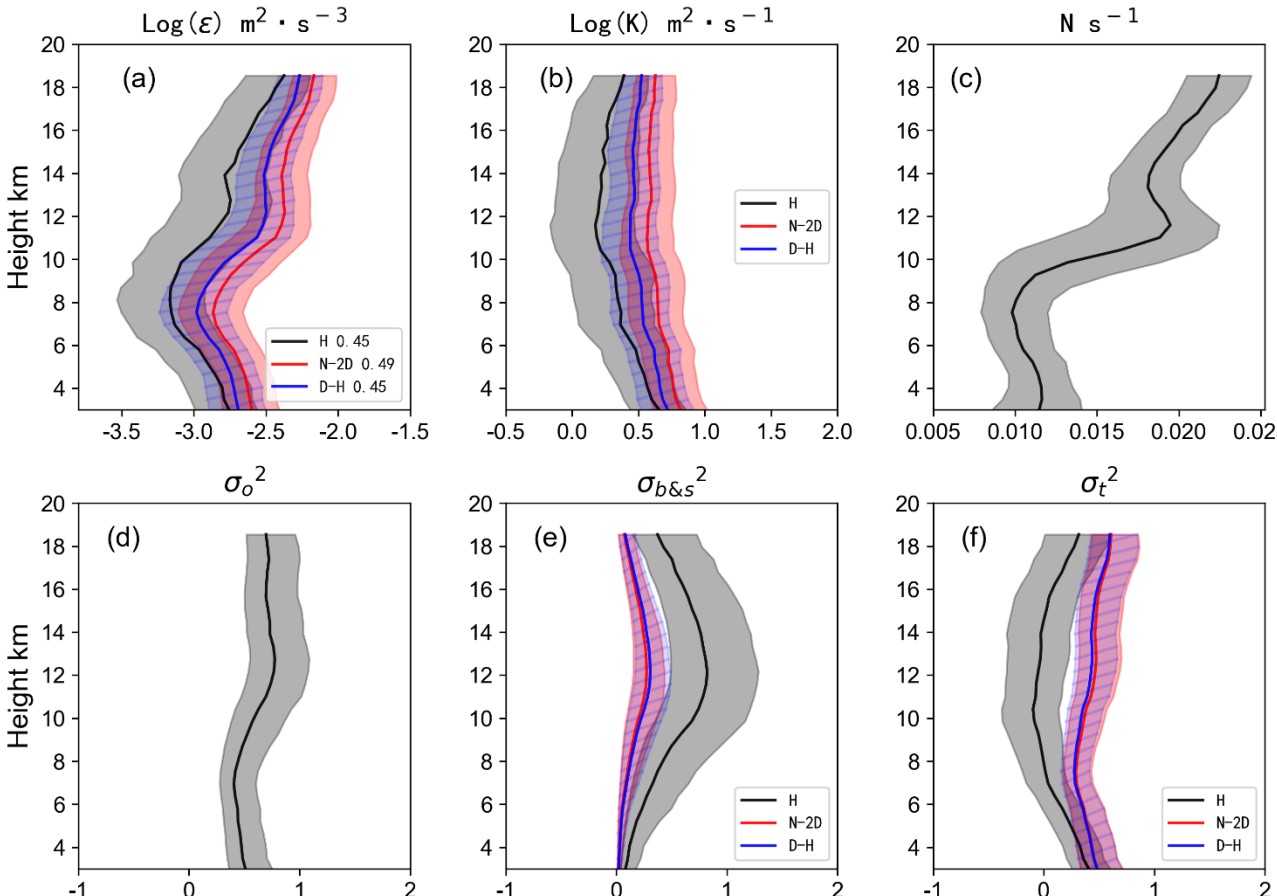

**Figure 5. Profiles of (a) $\varepsilon$, (b) Kz, (c) B-V frequency, (d) observation spectrum width, (e) beam and shear broadening, and (f) spectrum width caused by turbulence. The solid line is the median and the shaded area is the upper and lower quartiles. In panels**





**(a, b, e, f), the black/red/blue solid lines and shaded areas represent the median and upper and lower quartiles of the H model/N-2D model/D-H model, respectively.**

The distributional characteristics of the observed spectrum width calculated by Gaussian fitting are shown in Fig. 5(d).

The quartile of $\sigma_o^2$ (square of the Doppler velocity spectrum width) is between 0.2 and 1 m$^2$ s$^{-2}$. $\sigma_o^2$ increases with the altitude in the 7–13 km area, and $\sigma_o^2$ does not change much in the altitudinal range below 7 km and above 13 km. The B-V frequency is distributed between 0.01 and 0.025 s$^{-1}$, as shown in Fig. 5(c).

For the turbulent energy dissipation rate $\varepsilon$, the H model and D-H model have $c_1 = 0.45$, and the N-2D model has

$c_1 = A^{-\frac{3}{2}} \approx 0.49$ in this study. Figure 5(a, b) shows the average profiles of the turbulence parameter years $\varepsilon$ and Kz calculated

by the H model, N-2D model and D-H model. The distribution of $\sigma_t^2$ according to the N-2D model and D-H model is very consistent (Fig. 5f). Therefore, for facilitating comparison among the three models, we take the H model and D-H model have the same $c_1$.

Within the range of 3–19.8 km, there are differences in the $\varepsilon$ calculated by the three models, but there is good consistency in the trend of changes in height, as shown in Fig. 5(a). The $\varepsilon$ decreases with altitude from 3 km to 7 km, increases with altitude

from 7 km to 11 km, has no noticeable change from 11 km to 14 km, and decreases slowly with altitude above 14 km.

Using the H model, N-2D model and D-H model, the distribution ranges of the $\varepsilon$ quartiles are $10^{-3.5}$–$10^{-2.2}$ m$^2$ s$^{-3}$, $10^{-3.2}$–$10^{-2.1}$ m$^2$ s$^{-3}$ and $10^{-3.1}$–$10^{-2.0}$ m$^2$ s$^{-3}$, respectively. At an altitude of about 7 km, the $\varepsilon$ calculated by the three models reaches a minimum in each case, where the medians of the H model, N-2D model and D-H model are $10^{-3.2}$, $10^{-2.8}$ and $10^{-3.0}$ m$^2$ s$^{-3}$, respectively. In the range of 11–14 km, the medians of $\varepsilon$ in the H model, N-2D model and D-H model are $10^{-2.8}$, $10^{-2.4}$ and $10^{-2.5}$

m$^2$ s$^{-3}$, respectively.

It can be seen from Eq. (2) that the value of $c_1$ and the calculated turbulence spectrum width of different models have an impact on the calculation result of $\varepsilon$. The $\varepsilon$ calculated by the H model is significantly smaller than the calculated value of the other two models. The value of $c_1$ for the H model and D-H model is the same ($c_1$=0.45), but the turbulence spectrum width calculated by the H model is smaller than that of the D-H model at all heights [Fig. 5(f)]. The distributional characteristics of

the turbulence spectrum width at each height calculated by the D-H model and N-2D model are the same [Fig. 5(f)]. The value of $c_1$ in the N-2D model ($c_1$=0.49) is larger than that of the D-H model ($c_1$=0.45). As a result, the $\varepsilon$ calculated by the N-2D model is greater than the calculated value of the D-H model at each height. As shown in Fig. 5(e), the beam and shear broadening $\sigma_{b\&s}^2$ calculated by the H model are distributed discretely and are significantly larger than the calculation results of the other two models at each height. Therefore, the turbulence spectrum width calculated by the H model is the smallest and

relatively discrete.

For the vertical turbulence dissipation coefficient Kz, within the range of 3–19.8 km, the values of Kz calculated by the three models are different, but there is a good consistency with the changing trend of the height: Kz first decreases and then increases as the height increases. The minimum is at about 12 km. The medians of the Kz calculated by the H model, N-2D model and D-H model are respectively within $10^{0.18}$–$10^{0.67}$ m$^2$ s$^{-1}$, $10^{0.57}$–$10^{0.90}$ m$^2$ s$^{-1}$ and $10^{0.44}$–$10^{0.74}$ m$^2$ s$^{-1}$. The distributional

ranges of the upper and lower quartiles are $10^{-0.3}$–$10^{0.8}$ m$^2$ s$^{-1}$, $10^{0.4}$–$10^1$ m$^2$ s$^{-1}$ and $10^{0.2}$–$10^{0.8}$ m$^2$ s$^{-1}$, respectively.



## 4. Discussion

### 4.1 Applicability of the models in events

According to the probability of N-TKE in the three calculation models of the spectral width method, the applicability of
the three models under different conditions can be judged. Under the state that the N-TKE accounts for a relatively large
amount, the applicability of the corresponding model needs to be considered. For example, when the horizontal wind speed is
greater than 40 m s⁻¹, the proportion of N-TKE in the H model is greater than 60%. In the area above 7.5 km in Beijing, the
annual statistical results show that the N-TKE of the H model accounts for more than 40%. As shown in Fig. 6b, in the area
above 7.5 km in Beijing in July 2014, the probability of the N-TKE of the H model is relatively high, so the applicability of
the H model is lacking in this area.

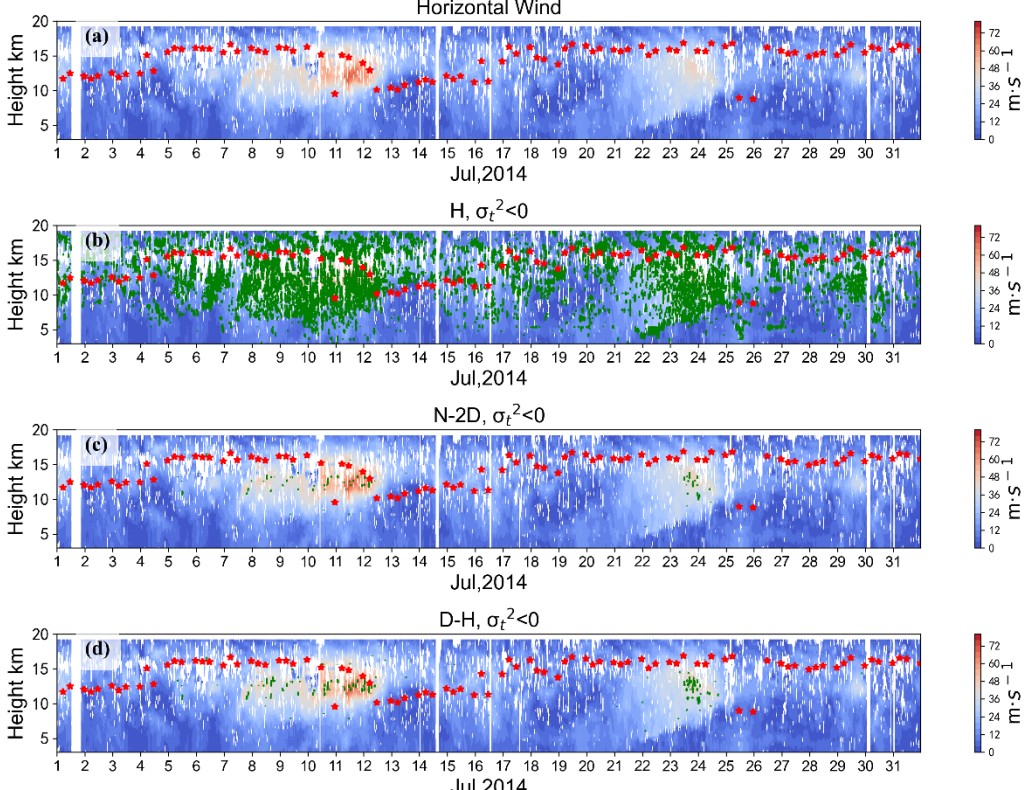

**Figure 6. N-TKE distribution of three models over the Beijing MST radar site in July 2014: (a) horizontal wind speed; (b–d) area of
N-TKE (green shading) for the (b) H model, (c) N-2D model and (d) D-H model. The red scattered points are the tropopause.**

Even when the statistical value of the probability of occurrence of the N-TKE of the model is low, the applicability of the
model still needs to be considered in some atmospheric processes. For example, for the N-2D and D-H models, when the
horizontal wind speed and the vertical shear of horizontal wind speed are within 0 to 60 m s⁻¹ and −0.02 to 0.02 s⁻¹, the rates
of N-TKE are less than 2% and 4%, respectively. Although as overall statistics the proportions of N-TKE being 2% and 4%



are very low, the values are higher in certain time period and height ranges, which is related to atmospheric processes and events indicated by the change of tropopause height. That is to say, we should pay more attention when dealing with the case
studies. It is also indicating the necessity to develop a universal model to calculate atmospheric turbulence parameters under the higher horizontal wind speed and vertical shear of horizontal wind speed circumstances. As shown in Fig. 6(c, d), there were two tropopause folding processes in the Beijing area in July 2014, and the horizontal wind speed was greater than 60 m s$^{-1}$ in the range of 10–15 km during 8–13 and 23–24 July. In the strong-wind area, the proportions of N-TKE in the N-2D model and DH model are higher. The results show that the N-2D model and D-H model, which have a relatively low rate of
N-TKE, still need to be modified to consider the model's applicability during the process of tropopause folding.

**4.2 Turbulence dissipation rate obtained using the middle and low mode**

The characteristics of the changes in $\varepsilon$ with height calculated by the mid-mode observational data of the Beijing MST radar agree well with existing research results. However, there is a difference in the range of values. The distributional characteristics of the median turbulence parameters of the Beijing MST radar are shown in Table 3.
**Table 3. Turbulence parameters of the Beijing MST radar (39.78°N, 116.95°E) at the range of 3–19.8 km.**

|  | H model, $c_1 = 0.45$ | N-2D model, $c_1 = 0.49$ | D-H model, $c_1 = 0.45$ |
|---|---|---|---|
| Median log ($\varepsilon$) (m$^2$ s$^{-3}$) | –3.2 (7 km) to –2.8 (14 km) | –2.8 (7 km) to –2.4 (14 km) | –3.0 (7 km) to –2.5 (14 km) |
| Median log (Kz) (m$^2$ s$^{-1}$) | 0.18–0.67 | 0.57–0.90 | 0.44–0.74 |

In addition to geographical differences, compared with other MST radars, the radial range resolution of the Beijing MST radar (600 m—other radars are generally 150 m) is the most different radar parameter. When using the spectral width method, it is necessary to satisfy the assumption that the observed atmospheric turbulence scale is smaller than the radar sampling
volume. To verify the impact of range resolution, we used the low-mode data (radial resolution of 150 m) of the Beijing MST radar to calculate turbulence parameters, and then compared them with the mid-mode results.

Based on the 3–7.8 km low-mode (mid-mode) data of the Beijing MST radar from 2012 to 2014, the H model, N-2D model and D-H model were applied, respectively. The results showed that the median $\varepsilon$ is $10^{-3.2}$ ($10^{-3.0}$) m$^2$ s$^{-3}$, $10^{-3.0}$ ($10^{-2.8}$) m$^2$ s$^{-3}$ and $10^{-3.1}$ ($10^{-2.9}$) m$^2$ s$^{-3}$ respectively. Also, the ratio of the median $\varepsilon$ of the middle and low mode is $10^{0.2}$ (approximately
1.6). The distributional characteristics of $\varepsilon$ obtained by applying the three models in the middle and low mode are basically the same, as shown in Fig. 7. The distribution of $\varepsilon$ obtained by the H model is between $10^{-5}$ and $10^{-1.5}$ m$^2$ s$^{-3}$, which is more discrete than the results of the other two models: the $\varepsilon$ obtained by the N-2D model and the D-H model is distributed between $10^{-4.5}$ and $10^{-1.5}$ m$^2$ s$^{-3}$. Compared with the middle mode, the $\varepsilon$ obtained by the three models has more of its distribution in the range of $10^{-2}$ to $10^{-1.5}$m$^2$ s$^{-3}$ in the low mode, indicating that the application of the low mode can detect stronger turbulence.





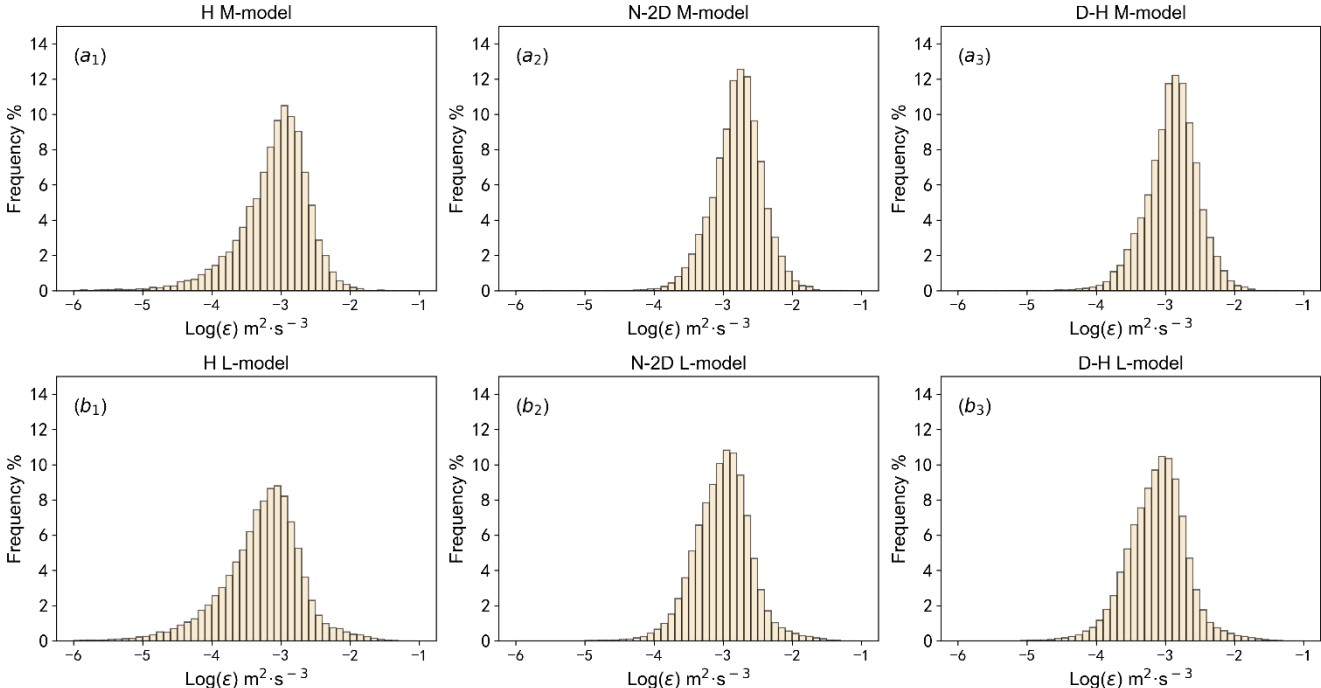

Figure 7. Distribution of $\varepsilon$ in the middle and low mode of the Beijing MST radar in the range of 3–7.8 km from 2012–2014: $(a_1 – a_3)$ distributional characteristics of $\varepsilon$ in the H model, N-2D model and D-H model (mid mode); $(b_1 – b_3)$ as in $(a_1 – a_3)$ but for low-mode data.

The Beijing MST radar and the Harrow VHF (very high frequency) radar (42.04°N) are at similar latitudes, and their

ranges of tropospheric $\varepsilon$ calculated by the H model show good consistency. The radial range resolution of the Harrow VHF

radar is 500 m, and the $\varepsilon$ is mainly distributed between $10^{-4}$ and $10^{-2}$ m$^2$ s$^{-3}$ in the altitudinal range of 1.5–11 km above the

radar site. There is also a certain proportion in the range of $10^{-5}$–$10^{-4}$ m$^2$ s$^{-3}$ and $10^{-2}$–$10^{-1.5}$ m$^2$ s$^{-3}$. The $\varepsilon$ calculated using the

ozone sounding (500–1000 m south of the Harrow radar) data is consistent with the radar calculation (Kantha and Hocking,

2011).

The above results show that the radial range resolution will affect the values of the turbulence parameters, but the effect

is relatively small. There are other reasons for the difference in turbulence parameters calculated by different radar data. For

example, when the dynamic stability is different, the value of $\varepsilon$ may be different. The gradient Richardson number (Ri) is a

dimensionless number used to judge dynamic stability. In Li et al. (2016), MAARSY radar (69.03° N, 16.04° E) data were

used to calculate $\varepsilon$, revealing that when Ri was < 1, the median $\varepsilon$ was $5.18 \times 10^{-4}$ m$^2$ s$^{-3}$ (W kg$^{-1}$), and when Ri was > 1, the

median ε was $1.61 \times 10^{-4}$ m$^2$ s$^{-3}$ (the former being 3.2 times that of the latter).



## 5. Conclusion

Based on the quality-controlled spectral width data of the Beijing MST radar from 2012 to 2014, including more than 37,000 profiles, three calculation models were used to calculate the turbulent spectral width. The turbulence parameters ($\varepsilon$, Kz)
over the station were calculated by the turbulent spectral width. Furthermore, the relationship between the proportion of N-TKE and both the domain of the horizontal wind speed and the vertical shear of horizontal wind was analyzed. The features of $\varepsilon$ using the mid- and low-mode observation models were compared, and the conclusions can be summarized as follows:

(1) The proportion of N-TKE in the H model, N-2D model and D-H model is sensitive to the horizontal wind field. The ratio of N-TKE in the H model increases with the horizontal wind speed and vertical shear of horizontal wind speed, up to
80%. When the horizontal wind speed is greater than 40 m s$^{-1}$, the proportion of N-TKE in the H model is greater than 60%, and the H model is not applicable. When the horizontal wind speed is greater than 20 m s$^{-1}$, the proportion of N-TKE in the N-2D model and D-H model only increases with the horizontal wind speed, and the maximum values are 2% and 4%, respectively. However, the applicability of the N-2D model and D-H model should be considered in some weather processes with strong winds, such as the process of tropopause folding.

(2) At all heights in Beijing, the horizontal wind speed in winter is greater than in summer. Therefore, the proportion of N-TKE at each height of the H model in winter is greater than that in summer, and the seasonal variation characteristics of the N-2D and D-H models at 6–11 km and 13–16 km are consistent with those of the H model. However, the N-2D and D-H models have no noticeable seasonal changes in the areas below 6 km and within 11–13 km in Beijing. The horizontal wind speed in the area below 6 km in the Beijing area is less than 20 m s$^{-1}$, and 11–13 km is the maximum horizontal wind speed
area. That is, the horizontal wind speed and vertical shear are close to 0 s$^{-1}$.

(3) Based on the observations of the Beijing MST radar in the altitudinal range of 3–19.8 km from 2012 to 2014, the median values of $\varepsilon$ in the H model, N-2D model and D-H model are $10^{-3.2}$–$10^{-2.8}$ m$^2$ s$^{-3}$, $10^{-2.8}$–$10^{-2.4}$ m$^2$ s$^{-3}$ and $10^{-3.0}$–$10^{-2.5}$ m$^2$ s$^{-3}$, respectively. The median values of Kz in the three models are $10^{0.18}$–$10^{0.67}$ m$^2$ s$^{-1}$, $10^{0.57}$–$10^{0.90}$ m$^2$ s$^{-1}$ and $10^{0.44}$–$10^{0.74}$ m$^2$ s$^{-1}$, respectively.

(4) Compared with previous studies, the turbulence parameters obtained by the three models in Beijing have the same variational trend with height. Still, there are differences in the distributional ranges of the turbulence parameters. Further analysis shows that different radial range resolutions of the radar have no apparent effect on the distributional ranges of the turbulence parameters.

When the spectral width method is used to calculate radar-based turbulence parameters, the statistical results in this paper
can provide a reference for the selection of the turbulence spectral width models. For example, when analyzing the statistical characteristics of the turbulence parameters over the radar station, a more suitable calculation model can be selected based on the local wind factors. The current results show that a more general model to calculate radar-based turbulence parameters should be proposed in researching the changes of turbulence parameters in specific weather processes.



## Data availability

Data related to this article are available upon request to the corresponding authors.

## Author contributions

Conceptualization: YFT, ZC, DRL; Data curation: ZC, YFT, DRL, YW; Formal analysis: ZC, YFT, DRL; Resources: YNW, YHB, XW, JH, LJP, YW; Visualization: ZC, YFT; Writing – original draft preparation: ZC, YFT, DRL; Writing – review & editing and scientific discussion and comments: ZC, YFT, DRL, YNW, YHB, XW, JH and LJP. DRL lead the team.

## Competing interests

The authors declare that they have no conflict of interest.

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
