# Peer review of "Turbulence parameters measured by the Beijing Mesosphere– Stratosphere–Troposphere radar in the troposphere/lower stratosphere with three models: Comparison and analyses"

_Atmospheric Measurement Techniques, 2021_

## Referee Comment (RC1)

Turbulence parameters measured by the Beijing MST radar

Ze Chen et al

This paper uses data from the Beijing MST radar to compare three methods of correcting the spectral width measured by the radar (at a zenith angle of 15°) for beam and shear broadening. It shows how the choice of method leads to a different likelihood of (apparent) negative turbulent broadening of the spectrum, depending on atmospheric conditions. The study is well constructed, the paper is clear and of value to the wind profiling community so I recommend publication subject to some minor corrections.

There is however a need to re-examine the calculations that have gone into this paper: see my comments on the actual formula for Nastrom and the value of $c_1$ for DH in the list below.

l.144 explain why specular reflection is a problem for calculating turbulence parameters. Also '…such as in a statically…'

Table 1 – explain what is meant by coherent and incoherent averaging (these terms don't mean much to readers unfamiliar with the MST radar technique)

l.156 How long a period is the data collected for a single observation (that is compared to the sonde)? 50 s?

l.186 broadening not widening

l 206 Equation 7 is wrong – it should read ….rcosχ) + $\theta^2/24$……  Also, having introduced the notation $\theta_{1/2}$ earlier, this expression should be used in this equation to avoid any confusion in the mind of the reader.

l.212 Deghan and Hocking's model is actually quite different to Nastrom's – it doesn't just contain one extra term for example. They derived it from their own 3-D simulations so you should explain more clearly how the two models differ.

General question – Dehghan and Hocking (paragraph leading to their eqn 10) argue that the factor 3 in the first term on the RHS of equation 7 is actually wrong and should be replaced by 4ln2 (2.77) if σ is the velocity variance, which is where their κ comes from.  Could you discuss this point and test what difference it makes to your work?

l.363 Why do you use $c_1 = 0.45$ for the DH model when Dehghan and Hocking themselves use 0.27? (just before their eqn 3)? Saying that you use 0.45 'to facilitate comparison' does not make sense.

---

## Referee Comment (RC2)

Comments on 'Turbulence parameters measured by the Beijing Mesosphere–Stratosphere–Troposphere radar in the troposphere/lower stratosphere with three models: Comparison and analyses' by Z. Chen et al.

**Summary**

In this study, three formulations for estimating broadening components of radar spectral widths proposed in previous studies are compared using three-year observations by a VHF radar at Xianghe Station, China. The debroadening algorithm is necessary for accurate estimation of turbulent energy dissipation rates because overestimation of broadening components will lead to negative energy dissipation rates. This study may include interesting results based on thorough comparison of broadening components and resulting energy dissipation rates, and I feel that the overall logic of the study is well organized. However, there seems to be an error in the calculation of the broadening component. Specifically, in the small vertical shear region, the three formulations should yield almost same results, but there is a significant difference between them. I suspect that the authors misunderstand the one-way and two-way beam width. I recommend the authors to recalculate them and reorganized the overall manuscript.

**General comments**

1. If there is no vertical shear ($|\partial \boldsymbol{u}/\partial z| = \sqrt{(\partial u/\partial z)^2 + (\partial v/\partial z)^2} = 0$), the three formulations are almost same:

Hocking (1985): $\qquad\qquad \sigma_b = \dfrac{\theta_{1/2}^{(2)} V}{\sqrt{2\ln 2}} = \dfrac{\theta_{1/2}^{(1)} V}{2\sqrt{\ln 2}},\ \ \sigma_s = 0,$

Nastrom (1997): $\qquad\qquad \sigma_{s\&b}^2 = \dfrac{\left(\theta_{1/2}^{(1)} V\right)^2}{3} \cos^2 \chi,$

Dehghan and Hocking (2011): $\qquad \sigma_{s\&b}^2 = \dfrac{\left(\theta_{1/2}^{(1)} V\right)^2}{4\ln 2} \cos^2 \chi,$

where $\theta_{1/2}^{(1)}$ and $\theta_{1/2}^{(2)}$ are one-way and two way half power half width of radar beam, respectively; $V$ is horizontal background wind; $\chi$ is the zenith angle of the beam (15°); and $4\ln 2 \approx 2.77$. Note that the formulations of Hocking (1985) and Dehghan and Hocking (2011) are identical in no shear condition except for $\cos^2 \chi$, which is approximately 0.93 (Table 1 in the manuscript). Figure 4 shows the vertical profiles of mean horizontal wind speed and vertical shear. It is found that mean vertical shear is very small around an altitude of 12.5 km, and thus the broadening components estimated from three formulations are expected to have similar values at the altitude. However, from Figure 5, $\sigma_{b\&s}^2$ from Hocking (1985)'s formulation is much larger than those from Nastrom (1997) and Dehghan and Hocking (2011). Other figures (e.g., Figure 2c, 3, 6, etc.) also show similar results. I am sure that something is wrong with the calculation of broadening components. One possibility is that the authors

are not aware of the difference between one-way and two-way half power half width ($\theta_{1/2}^{(1)} = \sqrt{2}\theta_{1/2}^{(2)}$).

2. P.16, L.349-352: I wonder why the difference of median energy dissipation rates between including negative values and excluding them is so small. In general, mean energy dissipation rates without negative values included will be quite large compared to that calculated from both positive and negative values. For example, Kurosaki et al. (1996) showed that median $K_z$ calculated from both positive and negative spectral widths are significantly different from that from only positive spectral width. Dehghan and Hocking (2011) showed a similar example for a radar with two-way beamwidth comparable to that used in this study. One of the exceptions is Kohma et al. (2019), who used an algorithm developed by Nishimura et al. (2020) to estimate the beam broadening component accurately. As a result, the difference of medians between with and without negative energy dissipation rates is small.

**Specific comments**
About notation: The notation for several variables is problematic in the present manuscript, e.g., $\theta$ is used as potential temperature (P. 3, L. 90) and half-power half width (P.9, L. 208), and both $\left|\frac{\partial u}{\partial z}\right|$ and $\left|\frac{\partial v}{\partial z}\right|$ are vertical gradient of horizontal wind. I recommend the authors to use consistent notations through the manuscript.

**Typos etc.**
Please specify the definition of vertical shear of horizontal wind:
Is it defined as $\sqrt{(\partial u/\partial z)^2 + (\partial v/\partial z)^2}$ ($u$ and $v$ is zonal and meridional wind, respectively)?

**References**

Dehghan, A., & Hocking, W. K. (2011). Instrumental errors in spectral‐width turbulence measurements by radars. Journal of Atmospheric and Solar-Terrestrial Physics, 73(9), 1052–1068. https://doi.org/10.1016/j.jastp.2010.11.011

Kohma, M., Sato, K., Tomikawa, Y., Nishimura, K., & Sato, T. (2019). Estimate of turbulent energy dissipation rate from the VHF Radar and radiosonde observations in the Antarctic. *Journal of Geophysical Research: Atmospheres*, *124*(6), 2976−2993. https://doi.org/10.1029/2018jd029521

Kurosaki, S., Yamanaka, M. D., Hashiguchi, H., Sato, T., & Fukao, S. (1996). Vertical eddy diffusivity in the lower and middle atmosphere :a climatology based on the MU radar observations during 1986-1992. Journal of Atmospheric and Terrestrial Physics, 58(6), 121-134.

Nastrom, G. D.: Doppler radar spectral width broadening due to beamwidth and wind shear, Annales Geophysicae, 15, 786-796, 10.1007/s00585-997-0786-7, 1997.

Nishimura, K., Kohma, M., Sato, K., & Sato, T. (2020). Spectral observation theory and beam debroadening algorithm for atmospheric radar. *IEEE Transactions on Geoscience and Remote Sensing*. https://doi.org/10.1109/TGRS.2020.2970200

---

## Author Comment (AC1)

**Cover letter**

**Dear Editors and Reviewers:**

Thank you for your letter and for the Reviewers' comments concerning our manuscript entitled "Turbulence parameters measured by the Beijing Mesosphere–Stratosphere–Troposphere radar in the troposphere/lower stratosphere with three models: Comparison and analyses" (ID: amt-2021-309). Those comments are valuable and helpful for revising and improving our paper and the important guiding significance to our research. We appreciate for Editors' and Reviewers' warm work earnestly and hope that the correction will meet with approval.

We have studied each comment carefully and have made corrections as far as possible. We recalculated the turbulence parameters and reorganized the overall manuscript, by taking $c_1$=0.27 for the D-H model and using one-way and two way half power half width of radar beam correctly. These changes do not affect the structure of the article. Revised portions are marked in red in the paper. For details of the revised manuscript, please refer to the document "Author's track-changes" and uploaded manuscript. The main corrections in the paper and the responses to the Reviewers' comments are as flowing:

**RC1 referee comment 1**

**Q1 l.144 explain why specular reflection is a problem for calculating turbulence parameters. Also '…such as in a statically…'**

Response:

Thank you very much for your comments and questions, the explaination is as follows:

Most research on turbulence parameters using atmospheric radar are based on the Kolmogorov hypothesis of isotropic turbulence at the inertial sub-region scale (Batchelor, 1953; Tatarski, 1961, 1971). To detect atmospheric turbulence intensity by atmospheric radars (spectral width method), the radar echo signal should come from turbulence scattering. In fact, at some heights, such as near the tropopause region, the radar echo can be affected by specular reflection. The specular (Fresnel) relection is not the same as that for refractive index perturbations due to turbulence. The layer can be treated as a plannar mirror with a small reflection for incident field. Therefore, if the vertical gradient of refractive index does not move vertically and does not change the shape of tratification either, the spectrum width will be affected by specular relection, making the spectral width method inapplicable.

The modified content revised version(clean) for Q1 is as follows:

Line No:146-149 (Section2.1, Page No:6 of 26)

"However, the vertical beam is more susceptible to specular reflection, especially in the tropopause region, which rather than isotropic scattering due to isotropic turbulence affects the spectrum (e.g., Fukao et al., 1994; Tsuda et al., 1986; Birner, 2006). The spectral width method is based on the isotropic scattering, which has the hypothesis that the radial wind speed variance (Doppler spectral width) detected by the radar is equal to the turbulence intensity."

**Q2 Table 1 – explain what is meant by coherent and incoherent averaging (these terms don't mean much to readers unfamiliar with the MST radar technique)**

Response:

Thank you very much for your suggestion.

The modified content revised version(clean) for Q2is as follows:

Coherent integration (uniting the signal phase at the identical range bin)

Incoherent integration (averaging processing in the frequency domain)

**Q3 l.156 How long a period is the data collected for a single observation (that is compared to the sonde)? 50 s?**

In medium mode, it takes about 5min for five beams to complete once data acquisition. For example, 2012.05.08, 15:40 (UTC). It starts at 35'4'' and ends at 40'10''. So it cost 5'6'' for five beams.

**Q4 l.186 broadening not widening**

Response:

Thank you for your reminding. We have replaced "Wind shear widening'' with "Wind shear broadening."

The modified content revised version(clean) for Q4 is as follows:

**Q5 l 206 Equation 7 is wrong – it should read ….rcosχ) + θ 2 /24…… Also, having introduced the notation θ½ earlier, this expression should be used in this equation to avoid any confusion in the mind of the reader**

Response:

Thank you very much for your comments and suggestions.

We have used the following expressions in the new version:

$\theta$ is used as potential temperature

$\theta_{1/2}^{(1)}$ is used as one way half power half width

$\theta_{1/2}^{(2)}$ is used as two way half power half width

$\frac{\partial u}{\partial z}$ is used as vertical gradient of horizontal wind

**Q6 l.212 Deghan and Hocking's model is actually quite different to Nastrom's – it doesn't just contain one extra term for example. They derived it from their own 3-D simulations so you should explain more clearly how the two models differ.**

Response:

Thank you for your reminder and suggestions.

Dehghan and Hocking (2011) gave a new calculation model (referred to as the D-H model) based on their own independent 3-D model as their reference, when Nastrom (1997) also introduced a 3-D model.

For Nastrom's (1997) model, the assumptions are as following:

The model made the following assumptions:

(i)      The model is two dimensional, and assumes only an xz vertical plane.

(ii)     The polar diagram is assumed to have sharp edges at 7n, where n is the one-way half-power width of the beam.

(iii)    The polar diagram is assumed to have constant gain within the region between n and +n, and changes abruptly to zero gain further from the beam center.

(iv)     All calculations are performed with the one-way beam only.

(v)      The pulse is assumed to be a square function.

(vi)     The returned power is calculated by assuming the pulse is centered at a fixed range R0, and does not consider that the returned power is a convolution between the scattering function and the pulse.

(vii)    No consideration is made for dependence of backscattered power on range within the pulse.

(viii)   The receiver is assumed to have an infinite beam-width.

(ix)     It is assumed that the beam-width is unchanged as the beam moves to off-zenith angles

Dehghan and Hocking (2011) allowed the various terms and cross-terms that they expected to have power-law formats, with the coefficients and power laws being left as variables. And they point out three important scales—the vertical projection of the pulse, the vertical distance from the lower point of the beam to the upper point, and the vertical projection of the distance for U to C. The three items can be covered by the main basic vertical length units, $\Delta r$ and $\zeta = 2r\theta_{1/2}^{(1)}sin\chi$. And they give the three-dimensional analytical expression for $\sigma_{s\&b}^2$.

To simplify the formula, the values for constant variables are given by the fitting. The final model formula is called model "c" by themselves, and the difference reduces to 5.7%. And Further improvement to mode C may be achieved by some modest adjustments at the critical point. They took the correction term involving incorporation of a set of Gaussian and hyperbolic tangent corrections, and the mean error reduced to 4.1%, relative to a full model.

"Dehghan and Hocking (2011) gave a new calculation model (referred to as the D-H model) based on their own independent 3-D model as their reference, while Nastrom (1997) also introduced a 3-D model. The simplified equation is as follows:"
* * *
**Q7**

**General question – Dehghan and Hocking (paragraph leading to their eqn 10) argue that the factor 3 in the first term on the RHS of equation 7 is actually wrong and should be replaced by 4ln2 (2.77) if σ is the velocity variance, which is where their κ comes from. Could you discuss this point and test what difference it makes to your work?**

**l.363 Why do you use c1 = 0.45 for the DH model when Dehghan and Hocking themselves use 0.27? (just before their eqn 3)? Saying that you use 0.45 'to facilitate comparison' does not make sense.**

Response:

As considered by the reviewer, the value of $c_1$ is very important, because it is significant for the calculation of turbulence parameters. But if the value of $c_1$ is constant, then only the values of $\varepsilon$ are affected, not the characteristic of $\varepsilon$ varying with height.

The distribution of $\sigma_t^2$ according to the N-2D model and D-H model is very consistent. And the trends with height of $\sigma_t^2$ are similar for the three models, when $\sigma_t^2$ calculated by the H model is smaller than that of the N-2D/ D-H model at all heights. The beam and shear broadening $\sigma_{b\&s}^2$ calculated by the H model are distributed discretely and are larger than the calculation results of the other two models at each height.

Obviously, the value of c1 is a critical issue that needs more research, but is not the focus of this paper. So the previous version of this paper considers the effect of spectral width, but this study does not discuss the effect of $c_1$ in-depth, so only two values of $c_1$ are selected to simplify things.

But as considered by the reviewers, the value of $c_1$ is poorly considered in the previous version. So, our new version takes $c_1$ as 0.27, as suggested by D-H.

As mentioned above, the specific influence of $c_1$ cannot be well explained in this work, but it is also the work to be carried out in the future.

The modified content revised version(clean) for Q7 is as follows:

We recalculated the turbulence parameters and reorganized the overall manuscript, by taking $c_1$=0.27 for D-H model

**RC2 referee comment 2**

In this study, three formulations for estimating broadening components of radar spectral widths proposed in previous studies are compared using three-year observations by a VHF radar at Xianghe Station, China. The debroadening algorithm is necessary for accurate estimation of turbulent energy dissipation rates because overestimation of broadening components will lead to negative energy dissipation rates. This study may include interesting results based on thorough comparison of broadening components and resulting energy dissipation rates, and I feel that the overall logic of the study is well organized. However, there seems to be an error in the calculation of the broadening component. Specifically, in the small vertical shear region, the three formulations should yield almost same results, but there is a significant difference between them. I suspect that the authors misunderstand the one-way and two-way beam width. I recommend the authors to recalculate them and reorganized the overall manuscript.

**General comments**

**Q1. If there is no vertical shear ($\left|\frac{\partial u}{\partial z}\right| = \sqrt{(\partial u_x/\partial z)^2 + (\partial v_y/\partial z)^2}$), the three formulations are almost same:**

**Hocking (1985):** $\sigma_b = \frac{\theta^{(2)}_{1/2} \cdot u}{\sqrt{2\ln 2}} = \frac{\theta^{(1)}_{1/2} \cdot u}{2\sqrt{\ln 2}}, \ \sigma_s = 0$

**Nastrom (1997):** $\sigma^2_{s\&b} = \frac{\theta^{(1)}_{1/2}{}^2}{3} u^2 \cos^2 \chi,$

**Dehghan and Hocking (2011):** $\sigma^2_{s\&b} = \frac{\theta^{(1)}_{1/2}{}^2}{k} u^2 \cos \chi,$

**where $\theta^{(1)}_{1/2}$ and $\theta^{(2)}_{1/2}$ are one-way and two way half power half width of radar beam, respectively; u is horizontal background wind; $\chi$ is the zenith angle of the beam (15°); and 4 ln 2 $\approx$2.77. Note that the formulations of Hocking (1985) and Dehghan and Hocking (2011) are identical in no shear condition except for $\cos\chi$, which is approximately 0.93 (Table 1 in the manuscript). Figure 4 shows the vertical profiles of mean horizontal wind speed and vertical shear. It is found that mean vertical shear is very small around an altitude of 12.5 km, and thus the broadening components estimated from three formulations are expected to have similar values at the altitude. However, from Figure 5, $\sigma^2_{s\&b}$ from Hocking (1985)'s formulation is much larger than those from Nastrom (1997) and Dehghan and Hocking (2011). Other figures (e.g., Figure 2c, 3, 6, etc.) also show similar results. I am sure that something is wrong with the calculation of broadening components. One possibility is that the authors are not aware of the difference between one-way and two-way half power**

**half width ($\theta_{1/2}^{(1)} = \sqrt{2}\theta_{1/2}^{(2)}$ )**

Thank you very much for your professional and specific comments and questions. We recalculate turbulence parameter and reorganized the overall manuscript, by using the one-way and two-way beam width correctly.

We re-analyzed the data and found that there were actually two problems:

1. One-way and two-way half power and half width are incorrectly understood.

2. Even if that mean vertical shear is very small (around an altitude of 12.5 km), $\sigma_{s\&b}^2$ calculated by H model is larger than the other two.

We will answer these two questions one by one.

**Question1. One-way and two-way half power and half width are incorrectly understood.**

We reviewed the code and found that H model used $\theta_{1/2}^{(1)}$ instead of $\theta_{1/2}^{(2)}$, so we rewrited the codes. And the comparison between before and after versions is shown in the figure below (Fig.1,2). $\theta_{1/2}^{(1)}$ is used as one way half power half width, $\theta_{1/2}^{(2)}$ is used as two way half power half width, and $\theta_{1/2}^{(1)} = \sqrt{2}\theta_{1/2}^{(2)}$.

The result shows that, compared with the previous version, $\sigma_{s\&b}^2$ calculated by the H model is smaller in the latest version. And the values ($\sigma_{s\&b}^2$, $\sigma_t^2$, $\varepsilon$, Kz) calculated by the three models are closer to each other.

But, the values of $\sigma_{s\&b}^2$ calculated by the H model is still larger than the other two models. Therefore, we conducted simulation experiments to verify the sensitivity of the three models to the vertical shear and horizontal wind speed. That's another question "2. Even if that mean vertical shear is very small (around an altitude of 12.5 km), $\sigma_{s\&b}^2$ calculated by H model is larger than the other two."

[Figure]

Figure 1 **Previous version**

Figure 2 Recalculated version (The latest version)

**2. Even if that mean vertical shear is very small (around an altitude of 12.5 km), $\sigma_{s\&b}^2$ calculated by H model is larger than the other two.**

The correct use of one way half power half width $\theta_{1/2}^{(1)}$ and two way half power half width $\theta_{1/2}^{(2)}$ reduce the differences of $\sigma_{s\&b}^2$ between the three models, especially around an altitude of 12.5 km (mean vertical shear is very small).

But in fact, the result is not as expected, the broadening components estimated from three formulations don't have similar values at the altitude (mean vertical shear is very small, around an altitude of 12.5 km). The values calculated by the H model are still larger than the other two models.

We conducted simulation experiments and further analyzed the cause. The reasons are as follows:

**(a)Simulation experiments:**

When $\frac{\partial u}{\partial z}$ is strictly equal to 0, the broadening components are slight different between the three models. But when $\frac{\partial u}{\partial z}$ is not strictly equal to 0, the H model will be significantly larger than the other two.

As shown in the figure, when take $\frac{\partial u}{\partial z}=\pm0.01$ s$^{-1}$(Fig.3 c, d), $\sigma_{s\&b}^2$ estimated from the H model is large than the other two models. One of the exceptions is that $\sigma_{s\&b}^2$ estimated from H model is similar to that from D-H model when $\frac{\partial u}{\partial z}=$-0.01 s$^{-1}$.

[Figure]

**Figure 3** The $\sigma^2_{s\&b}$ estimated from the three models relate to the vertical shear and horizontal wind speed. (a)(b) $\frac{\partial u}{\partial z}$

is in the range of -0.02 to 0.02 s$^{-1}$, per 0.002 s$^{-1}$. (c)(d) $\frac{\partial u}{\partial z}$ has just 3 numbers, -0.01 s$^{-1}$, 0 s$^{-1}$, 0.01 s$^{-1}$. (a)(c) are

$\sigma^2_{s\&b}$, (b)(d) are log ($\sigma^2_{s\&b}$). (c)(d) red lines are $\frac{\partial u}{\partial z}$=-0.01 s$^{-1}$, blue lines are $\frac{\partial u}{\partial z}$=0 s$^{-1}$, purple lines are $\frac{\partial u}{\partial z}$=-0.01

s$^{-1}$. (c) The stars are $\frac{\partial u}{\partial z} = 0$ s$^{-1}$. The broadening components estimated from H model is just related to $|\frac{\partial u}{\partial z}|$,

so there are just 5 lines instead of 6 lines.

**(b) Seasonal characteristics of $\frac{\partial u}{\partial z}$ in Beijing**

1. In fact, the height of maximum horizontal wind speed has obvious seasonal variation in Beijing. The positive

and negative values have an impact on the statistical results of $\frac{\partial u}{\partial z}$. This leads to mean vertical shear being very

small at 12.5 km in Beijing. The median, upper and lower quartile profiles of $|\frac{\partial u}{\partial z}|$ are shown in Fig.4.

2. There are always strong vertical shear for several kilometers above ($\frac{\partial u}{\partial z} < 0$) or below ($\frac{\partial u}{\partial z} > 0$) the height of

maximum horizontal wind speed.

[Figure]

Figure 4 The median, upper and lower quartile profiles of $|\frac{\partial u}{\partial z}|$. It is the same as Fig4 in the manuscript, but for absolute value of $\frac{\partial u}{\partial z}$.

To sum up, although that the mean vertical shear is very small around an altitude of 12.5 km in Beijing, the horizontal wind vertical shear at this height is not strictly equal to 0 for many individual profiles. As the Simulation experiments show, when $\frac{\partial u}{\partial z}$ is not strictly equal to 0, the turbulence spectrum width of H model will be larger than that of the other two models. Thus it is reasonable that not all of them, the broadening components estimated from three formulations, have similar values at the altitude (mean vertical shear is very small).

The modified content revised version(clean) for Q1 is as follows:

We recalculate turbulence parameter and reorganized the overall manuscript, by using the one-way and two-way beam width correctly.

**Q2: P.16, L.349-352: I wonder why the difference of median energy dissipation rates between including negative values and excluding them is so small. In general, mean energy dissipation rates without negative values included will be quite large compared to that calculated from both positive and negative values. For example, Kurosaki et al. (1996) showed that median $Kz$ calculated from both positive and negative spectral widths are significantly different from that from only positive spectral width. Dehghan and Hocking (2011) showed a similar example for a radar with two-way beamwidth comparable to that used in this study. One of the exceptions is Kohma et al. (2019), who used an algorithm developed by Nishimura et al. (2020) to estimate the beam broadening component accurately. As a result, the difference of medians between with and without**

**negative energy dissipation rates is small.**

Response:

Thank you very much for your comments and questions. We use log ($\varepsilon$)/ log (Kz) for statistics and give the average of the median of each height. Such expressions may mislead the reader.

Therefore, we make statistics on $\varepsilon$/Kz and log ($\varepsilon$)/ log (Kz) after recalculating the turbulence parameters. And the statistics are as follows (Table1):

**Table 1 The ratios of the median mean $\varepsilon$ and Kz calculated by the H model, N-2D model and D-H model (including/excluding negative values)**

|  | including/excluding negative values | H (C1=0.45) | N-2D(C1=0.49) | D-H (C1=0.27) |
|---|---|---|---|---|
| $\varepsilon$ | including negative values | 0.0019 | 0.0034 | 0.0018 |
|  | excluding negative values | 0.0029 | 0.0034 | 0.0020 |
| Kz | including negative values | 2.39 | 4.37 | 2.24 |
|  | excluding negative values | 3.66 | 4.37 | 2.47 |
| Log($\varepsilon$) | including negative values | -2.7 | -2.5 | -2.8 |
|  | excluding negative values | -2.6 | -2.5 | -2.8 |
| Log(Kz) | including negative values | 0.48 | 0.64 | 0.35 |
|  | excluding negative values | 0.56 | 0.64 | 0.39 |

The profiles of $\varepsilon$, Kz, log ($\varepsilon$) and log (Kz) are shown in Figures 5 to 8。

The result shows that the difference between including negative values and excluding them is closely related to the proportion of N-TKE. Compared with the H model, the differences between including negative values and excluding them are very small for the N-2D model and D-H model, which is because the H model has a higher proportion of N-TKE.

The modified content revised version(clean) for Q2is as follows:

Line No:373-382 (Section3.3, Page No:17-18 of 26)

"The turbulence spectrum width contains negative values, as do $\varepsilon$ and Kz. The difference between including negative values and excluding them is closely related to the proportion of N-TKE. Compared with H model, the difference between including negative values and excluding them is very small for N-2D model and D-H model, which is due to the fact that the H model has a higher proportion of N-TKE. The ratios of the median mean $\varepsilon$ calculated by the H model, N-2D model and D-H model (including/excluding negative values) are 0.019/0.029, 0.0034/0.0034 and 0.0018/0.0020, respectively. The ratio of Kz is 2.39/3.66, 4.37/4.37 and 2.24/2.47, respectively. Several studies

showed that the mean energy dissipation rates without negative values included will be quite large compared to that calculated from both positive and negative values (Kurosaki et al. 1996; Dehghan and Hocking, 2011). One of the exceptions is Kohma et al. (2019), who used an algorithm developed by Nishimura et al. (2020) to estimate the beam broadening component accurately. As a result, the difference of medians between with and without negative energy dissipation rates is small."

[Figure]

Figure 5 log ($\varepsilon$) and log (Kz) including negative values.

[Figure]

Figure 6 log ($\varepsilon$) and log (Kz) excluding negative values.

[Figure]

Figure 7 $\varepsilon$, Kz including negative values.

[Figure]

Figure 8 $\varepsilon$, Kz excluding negative values.

**Q3:**
**Specific comments**
**About notation: The notation for several variables is problematic in the present manuscript, e.g, $\theta$ is used as potential temperature (P.3, L.90) and half-power half width (P.0, L.208), and both $|\frac{\partial u}{\partial z}|$ and $|\frac{\partial v}{\partial z}|$ are vertical gradient of horizontal wind. I recommend the authors to use consistent notations through the manuscript.**

Response:

Thank you very much for your comments and suggestions.

We have used consistent notation through the manuscript in the latest version as follows:

$\theta$ is used as potential temperature

$\theta_{1/2}^{(1)}$ is used as one way half power half width

$\theta_{1/2}^{(2)}$ is used as two way half power half width

$\frac{\partial u}{\partial z}$ is used as vertical gradient of horizontal wind

**Q4:**
**Typos etc.**
**Please specify the detinition of vertical shear of horizontal wind:**
**Is it defined as $\sqrt{\left(\frac{\partial u}{\partial z}\right)^2 + \left(\frac{\partial v}{\partial z}\right)^2}$ ($u$ and $v$ is zonal and meridional wind, respectively)?**

Response:

Thank you very much for your suggestions. As you have considered, $\frac{\partial u}{\partial z}$ is a question worth considering.

We assume that, i. The wind profile is linear. ii. Within a sampling volume of two range bins, regardless of the influence of wind direction. So values of $\frac{\partial u}{\partial z}$ can take into account the change in scalar values.

However, the horizontal wind we get is a vector, and there must be changes in wind directions.

In these three models, $\frac{\partial u}{\partial z}$ is used to calculate the horizontal wind speed at each point in the sample volume and $\frac{\partial u}{\partial z}$ is assumed to be constant.

But all three models are simplified, so we cannot be sure that a scalar or vector representation is more reasonable.

One point is inevitable, $\frac{\partial u}{\partial z}$ has positive and negative.

Scalar:

$$\partial u / \partial z_s = \frac{\partial \sqrt{u_x{}^2 + v_y{}^2}}{\partial z}$$

About vector, we think the following formula is more suitable:

$$\partial u / \partial z_v = \begin{cases} (-1) \cdot \sqrt{\left(\frac{\partial u_x}{\partial z}\right)^2 + \left(\frac{\partial v_y}{\partial z}\right)^2}, & \frac{\partial \sqrt{u_x{}^2 + v_y{}^2}}{\partial z} < 0 \\ 1 \cdot \sqrt{\left(\frac{\partial u_x}{\partial z}\right)^2 + \left(\frac{\partial v_y}{\partial z}\right)^2}, & \frac{\partial \sqrt{u_x{}^2 + v_y{}^2}}{\partial z} > 0 \end{cases}$$

where $u_x$ and $v_y$ is zonal and meridional wind, respectively.

So we use the observational data from Beijing MST (2012 to 2014) to analyze the differences between $\partial u / \partial z_s$ and $\partial u / \partial z_v$ as shown in Fig.9. The result shows there is little difference between them, the mean absolute value of $(\partial u / \partial z_s - \partial u / \partial z_v)$ is 0.00047. And we take $\partial u / \partial z \equiv \partial u / \partial z_v$ in this study.

[Figure]

Figure 9 Difference of $\frac{\partial u}{\partial z}$ between scalar and vector

The modified content revised version(clean) for Q4 is as follows:

Line No:196-208 (Section2.3.1, Page No:8-9 of 26)]

"In fact, the horizontal wind we get is a vector, and there must be changes in wind directions. In these three models, $\frac{\partial u}{\partial z}$ is used to calculate the horizontal wind speed at each point in the sample volume and is assumed to be constant. But all three models are simplified, so we cannot be sure that a scalar or vector representation is more reasonable. One point is inevitable, $\frac{\partial u}{\partial z}$ has positive and negative values.

About scalar, the vertical shear of horizontal wind $\partial u / \partial z_s$ is as following:

$$\partial u / \partial z_s = \frac{\partial \sqrt{u_x^2 + v_y^2}}{\partial z},$$ (6)

About vector, we think the following formula is more suitable:

$$\partial u / \partial z_v = \begin{cases} (-1) \cdot \sqrt{\left(\frac{\partial u_x}{\partial z}\right)^2 + \left(\frac{\partial v_y}{\partial z}\right)^2}, & \frac{\partial \sqrt{u_x^2 + v_y^2}}{\partial z} < 0 \\ 1 \cdot \sqrt{\left(\frac{\partial u_x}{\partial z}\right)^2 + \left(\frac{\partial v_y}{\partial z}\right)^2}, & \frac{\partial \sqrt{u_x^2 + v_y^2}}{\partial z} > 0 \end{cases},$$ (7)

where $u_x$ and $v_y$ is zonal and meridional wind, respectively.

So we use the observational data from Beijing MST (2012 to 2014) to analyze the differences between $\partial u / \partial z_s$ and $\partial u / \partial z_v$. The result shows there is little difference between them, and the mean absolute value of $\partial u / \partial z_s$-$\partial u / \partial z_v$ is 0.00047. We take $\partial u / \partial z \equiv \partial u / \partial z_v$ in this study."

---

## Referee Report (RR1)

Turbulence parameters measured by the Beijing MST radar

Ze Chen et al

The authors have addressed my concerns with the original version but in making their changes they have introduced some further points that need addressing.

l.100 is related to (not with)

Table 1. Coherent integration (combining signals from the same height bin over successive radar pulses, according to phase)

Incoherent integration (averaging of spectra)

Add to the table or the following text how long it takes to make one observation with the radar (your reply to me stated 5'6'' but you didn't put that in the paper)

l.156-7 '… tropopause region, where the echo signal spectrum is narrow and unrelated to turbulence (e.g Fukao…..'

l.158 based on isotropic scattering

l.210-230. When thinking about the physics of the way wind shear affects the spectral width, we must recall that the Doppler shift is due to the component of velocity along the beam direction. If the beam is tilted along the same azimuth as the wind, and there is a vertical wind shear in the same direction, the spectrum is affected as shown in Dehgan and Hocking (2011), fig 5a. For example, positive wind shear makes the Doppler shift of the 'top' of the beam larger relative to the 'bottom', reducing the difference between them. But if the wind rotates with height in the range gate (transverse wind shear), the extra wind components add in quadrature to the mean wind, and their impact on the wind speed is much smaller. So the correct value of wind shear should be $\partial u \partial z|_{\phi}$ where $\phi$ is the azimuth direction of the mean wind.

---

## Referee Report (RR2)

Comments on 'Turbulence parameters measured by the Beijing Mesosphere–Stratosphere–Troposphere radar in the troposphere/lower stratosphere with three models: Comparison and analyses' by Z. Chen et al.

**Summary**

The reviewer acknowledges the replies by the authors. In my opinion, the study fits into the scope of Atmospheric Measurement Techniques, while I have several comments which the authors might want to address in further revised version.

**Specific comments**

In the revised manuscript, the authors added a discussion on how to calculate the vertical shear of horizontal wind for estimating the shear broadening component, where the vertical shear is calculated as the vertical gradient of absolute value of horizontal wind vector. However, since shear broadening comes from the variation of the radial velocity within the radar volume, only the beam direction component of the horizontal wind vector contributes to the broadening of the radar spectrum. In fact, Nastrom (1997) shows scatter plots of spectral widths of the eastward (northward) beam versus vertical shear of zonal (meridional) winds to explore the shear broadening effects of WSMR observations. I recommend the authors to recalculate shear broadening effects.

**References**

Nastrom, G. D.: Doppler radar spectral width broadening due to beamwidth and wind shear, Annales Geophysicae, 15, 786-796, 10.1007/s00585-997-0786-7, 1997.

---

## Author Response (AR2)

**Cover letter**

**Dear Editors and Reviewers:**

Thank you for your letter and for the Reviewers' comments concerning our manuscript entitled "Turbulence parameters measured by the Beijing Mesosphere–Stratosphere–Troposphere radar in the

5    troposphere/lower stratosphere with three models: Comparison and analyses" (ID: amt-2021-309). Those comments are valuable and helpful for revising and improving our paper and the important guiding significance to our research. We appreciate for Editors' and Reviewers' warm work earnestly and hope that the correction will meet with approval.

We have studied each comment carefully and have made corrections as far as possible. We recalculated

10    the turbulence parameters and reorganized the overall manuscript, by taking the correct value of wind shear as $\frac{\partial u}{\partial z}_\phi$, where $\phi$ is the azimuth direction of the mean wind. In the revised version, we take the zonal (meridional) winds to explore the shear broadening effects of the east and west (north and south) beam. These changes do not affect the structure of the article. Revised portions are marked in red in the paper. For details of the revised manuscript, please refer to the document "Author's track-changes"

15    and uploaded manuscript. The main corrections in the paper and the responses to the Reviewers' comments are as flowing:

**List of Response**

20   **RC1 referee comment 1**

**Q1 l.100 is related to (not with)**

Response:

Thank you very much for your suggestion.

The modified content revised version(clean) for Q1 is as follows:

25   Line No:92 (Section 1, Page No:3 of 22)

"That is, several studies pointed out that the velocity variance measured by the radar is related to the transverse one-dimensional spectrum function for the direction radial from the radar (Dehghan &Hocking, 2011; Hocking, 1999)."

30   **Q2 Table 1 – Coherent integration (combining signals from the same height bin over successive radar pulses, according to phase)**

**Incoherent integration (averaging of spectra)**

**Add to the table or the following text how long it takes to make one observation with the radar (your reply to me stated 5'6" but you didn't put that in the paper)**

35   Response:

Thank you very much for your suggestion.

We have Added to the following text how long it takes to make one observation with the radar.

The modified content revised version(clean) for Q2 is as follows:

Line No:143 (Section 2.1, Page No:4 of 22)

Coherent integration (combining signals from the same height bin over successive radar pulses, according to phase)

Incoherent integration (averaging of spectra )

40   Line No:141-142 (Section 2.1, Page No:4 of 22)

"In middle mode, it takes about 5min for five beams to complete once data acquisition."

**Q3 l.156-7 '… tropopause region, where the echo signal spectrum is narrow and unrelated to turbulence (e.g Fukao…..'**

**l.158 based on isotropic scattering**

Respnse:

Thank you very much for your suggestion.

The modified content revised version(clean) for Q3 is as follows:

Line No:148‐150 (Section 3.1, Page No:5 of 22)

"However, the vertical beam is more susceptible to specular reflection, especially in the tropopause region, where the echo signal spectrum is narrow and unrelated to turbulence (e.g., Fukao et al., 1994; Tsuda et al., 1986; Birner, 2006), which based on isotropic scattering."

**Q4 l.210-230. When thinking about the physics of the way wind shear affects the spectral width, we must recall that the Doppler shift is due to the component of velocity along the beam direction. If the beam is tilted along the same azimuth as the wind, and there is a vertical wind shear in the same direction, the spectrum is affected as shown in Dehgan and Hocking (2011), fig 5a. For example, positive wind shear makes the Doppler shift of the 'top' of the beam larger relative to the 'bottom', reducing the difference between them. But if the wind rotates with height in the range gate (transverse wind shear), the extra wind components add in quadrature to the mean wind, and their impact on the wind speed is much smaller. So the correct value of wind shear should be $\frac{\partial u}{\partial z}_\phi$, where $\phi$ is the azimuth direction of the mean wind.**

Response:

Thank you very much for your reminder and suggestions.

As you may have considered, we have considered the issue of "component of the horizontal wind" in the new version, and recalculated shear broadening effects.

There are two main jobs, 1. The preparations of calculating shear broadening effects. 2. The selection of statistical samples.

**1. Preparations**

**This study used the the absolute value of the component of the horizontal wind vector, did not overdiscuss the effect of wind direction, where $\frac{\partial u}{\partial z}$ contains positive and negative values.**

**1.1 The beam direction component of the horizontal wind**

Only the beam direction component of the horizontal wind vector contributes to the broadening of the radar spectrum. So the correct value of wind shear should be $\frac{\partial u}{\partial z}_\phi$, where $\phi$ is the azimuth direction of the mean wind (Nastrom, 1997; Dehgan and Hocking, 2011). In this study, we take the zonal (meridional) winds to explore the shear broadening effects of the east and west (north and south) beam. The vertical shear of horizontal wind $\frac{\partial u}{\partial z}$ is as following:

For the east and west beams:

$$u = u_x, \frac{\partial u}{\partial z} = \frac{\partial u_x}{\partial z} \tag{6}$$

For the north and south beams:

$$u = v_y, \frac{\partial u}{\partial z} = \frac{\partial v_y}{\partial z} \tag{7}$$

where $u_x$ and $v_y$ is zonal and meridional wind, respectively.

**1.2 The directions of $u_x$ and $v_y$**

The models take the u as the horizontal wind speed(Nastrom, 1997; Dehgan and Hocking, 2011). In fact, the direction of the component of the horizontal wind is indicated by a positive or negative number, such as th positive value of $u_x$ means the wind blows from west to east. So we do some simulation experiments for these three models, the results show that the directions of $u_x$ and $v_y$ have no effect on the results of H model, and have very little effect on D-H model and N-2D model, as shown in Fig.R-1.

**This study used the the absolute value of the component of the horizontal wind vector, did not overdiscuss the effect of wind direction, where $\frac{\partial u}{\partial z}$ contains positive and negative values.**

**Simulation experiments:**

[Figure]

**Figure R-1** The $\sigma_{s\&b}^2$ estimated from the three models relate to the vertical shear and horizontal wind speed. The horizontal wind speed is in the range of -80 to 80 m/s, per 4m/s. (a)(b) $\frac{\partial u}{\partial z}$ is in the range of -0.02 to 0.02 s⁻¹, per 0.002 s⁻¹. (c)(d) $\frac{\partial u}{\partial z}$ has just 3 numbers, -0.01 s⁻¹, 0 s⁻¹, 0.01 s⁻¹. (a)(c) are $\sigma_{s\&b}^2$, (b)(d) are log ($\sigma_{s\&b}^2$). (c)(d) red lines are $\frac{\partial u}{\partial z}$=-0.01 s⁻¹, blue lines are $\frac{\partial u}{\partial z}$=0 s⁻¹, purple lines are $\frac{\partial u}{\partial z}$=0.01 s⁻¹. (c) The stars are $\frac{\partial u}{\partial z} = 0$ s⁻¹. The broadening components estimated from H model is just related to $|\frac{\partial u}{\partial z}|$, so it is the same line for H model taking $\frac{\partial u}{\partial z}$ as 0.01 s⁻¹ or -0.01 s⁻¹.

The modified content revised version(clean) for calculating shear broadening effects is as follows: Line No:198-208 (Section 2.3.1, Page No:6 of 22)

"In fact, only the beam direction component of the horizontal wind vector contributes to the

broadening of the radar spectrum. So the correct value of wind shear should be $\frac{\partial u}{\partial z}_\phi$, where $\phi$ is the azimuth direction of the mean wind (Nastrom, 1997; Dehgan and Hocking, 2011). In this study, we take the zonal (meridional) winds to explore the shear broadening effects of the east and west (north and south) beam. The vertical shear of horizontal wind $\frac{\partial u}{\partial z}$ is as following:

110     For the east and west beams:

$$u = u_x, \frac{\partial u}{\partial z} = \frac{\partial u_x}{\partial z} \tag{6}$$

    For the north and south beams:

$$u = v_y, \frac{\partial u}{\partial z} = \frac{\partial v_y}{\partial z} \tag{7}$$

where $u_x$ and $v_y$ is zonal and meridional wind, respectively. And the directions of $u_x$ and $v_y$ have

115 no effect on the results of H model, and have very little effect on D-H model and N-2D model. This study used the the absolute value of the component of the horizontal wind vector, did not overdiscuss the effect of wind direction, where $\frac{\partial u}{\partial z}$ contains positive and negative values."

**2. The statistical samples – four oblique beams**

120 The "**Preparations**" showed clearly how to recalculated shear broadening effects. But how to choose the the statistical samples is another thing that needs careful consideration. We have the observational datasets of four oblique beams, but there are obvious variation in the distributions of $u_x$ and $v_y$ (as shown in Fig. R-2). here $u_x$ and $v_y$ is zonal and meridional wind, respectively.

[Figure]

Figure R-2. Two-dimensional frequency distribution characteristics of horizontal wind speed and vertical shear of horizontal wind speed within the height range of 3–19.8km above the Beijing MST radar station from 2012 to 2014. (a)(b)(c) The east-west component of horizontal wind, (d)(e)(f) The north-south component of horizontal wind.

The east-west component of the horizontal wind speed over the radar site is distributed between 0 m s$^{-1}$ and 60 m s$^{-1}$, and the vertical shear of the horizontal wind speed ranges from −0.014 to 0.014 s$^{-1}$. The north-south component of the horizontal wind speed in Beijing is distributed between 0 m s$^{-1}$ and 20 m s$^{-1}$, and the vertical shear of the horizontal wind speed ranges from −0.014 to 0.014 s$^{-1}$.

As shown in Table R-1, for the east and west beams, the rates of N-TKE ($\sigma_t^2 < 0$) of the H model, N-2D model and D-H model are in the range of 27%–32%, 15%–21% and 9%–15%, respectively. And for the north and south beams, the rates are in the range of 5%–8%, 2%–4% and 0.6%–1.0%,. The probability that the turbulence spectrum width is less than 0 calculated by different oblique beams are different. But results of the symmetric beams (such as east and west beams/ north and south beams) are similar.

140      Table R-1. Total frequency of $\sigma_t^2 < 0$ in the range of 3–19.8 km.

| Beams | Time | Total numbers | H, $\sigma_t^2 < 0$ | N-2D, $\sigma_t^2 < 0$ | D-H, $\sigma_t^2 < 0$ |
|---|---|---|---|---|---|
| East | 2012 | 287490 | 78484 (27.30%) | 43253(15.05%) | 28067(9.76%) |
| | 2013 | 278317 | 76038(27.32%) | 43886(15.77%) | 27836(10.00%) |
| | 2014 | 311233 | 90633(29.12%) | 54988(17.67%) | 34219(10.99%) |
| West | 2012 | 288060 | 82821(28.75%) | 46925(16.29%) | 32467(11.27%) |
| | 2013 | 280769 | 82019(29.21%) | 48156(17.15%) | 32931(11.73%) |
| | 2014 | 313848 | 103226(32.89%) | 64997(20.71%) | 44683(14.24%) |
| North | 2012 | 102079 | 7924(7.76%) | 3870(3.79%) | 923(0.90%) |
| | 2013 | 84402 | 6377(7.56%) | 3206(3.81%) | 724(0.86%) |
| | 2014 | 92084 | 5900(6.41%) | 3115(3.38%) | 726(0.79%) |
| South | 2012 | 101288 | 6932(6.84%) | 3583(3.54%) | 985(0.97%) |
| | 2013 | 83418 | 5635(6.76%) | 2985(3.58%) | 696(0.83%) |
| | 2014 | 91535 | 5061(5.52%) | 2674(2.92%) | 573(0.63%) |

So if the distributions of $u_x$ and $v_y$ won't affect the statistical results, we take the observational datasets of four oblique beams as a total sample. Otherwise, we take the datasets of east and west

145    beams as one sample and that of north and south beams as the other sample.

The modified content revised version(clean) for **The statistical samples** is as follows:

150

**For Tabel 2. We take four samples for every year.**

"

Table 1. Total frequency of $\sigma_t^2 < 0$ in the range of 3–19.8 km.

| Beams | Time | Total numbers | H, $\sigma_t^2 < 0$ | N-2D, $\sigma_t^2 < 0$ | D-H, $\sigma_t^2 < 0$ |
|---|---|---|---|---|---|
| East | 2012 | 287490 | 78484 (27.30%) | 43253(15.05%) | 28067(9.76%) |
| | 2013 | 278317 | 76038(27.32%) | 43886(15.77%) | 27836(10.00%) |
| | 2014 | 311233 | 90633(29.12%) | 54988(17.67%) | 34219(10.99%) |
| West | 2012 | 288060 | 82821(28.75%) | 46925(16.29%) | 32467(11.27%) |
| | 2013 | 280769 | 82019(29.21%) | 48156(17.15%) | 32931(11.73%) |
| | 2014 | 313848 | 103226(32.89%) | 64997(20.71%) | 44683(14.24%) |
| North | 2012 | 102079 | 7924(7.76%) | 3870(3.79%) | 923(0.90%) |
| | 2013 | 84402 | 6377(7.56%) | 3206(3.81%) | 724(0.86%) |
| | 2014 | 92084 | 5900(6.41%) | 3115(3.38%) | 726(0.79%) |
| South | 2012 | 101288 | 6932(6.84%) | 3583(3.54%) | 985(0.97%) |
| | 2013 | 83418 | 5635(6.76%) | 2985(3.58%) | 696(0.83%) |
| | 2014 | 91535 | 5061(5.52%) | 2674(2.92%) | 573(0.63%) |

"

155

**For Fig1, We take the datasets of east and west beams as one sample and that of north and south beams as the other sample.**

160  "

[Figure]

Figure 3. Two-dimensional frequency distribution characteristics of horizontal wind speed and vertical shear of horizontal wind speed within the height range of 3–19.8km above the Beijing MST radar station from 2012 to 2014. (a)(b)(c) The east-west component of horizontal wind, (d)(e)(f) The north-

165  south component of horizontal wind.

"

**For Fig2 This paper just gives the resultes of the east-west component of horizontal wind. Beacause the north-south component of horizontal wind speed in Beijing is distributed between 0 m s⁻¹ and 20 m s⁻¹.**

"

Figure 4. Frequency distribution of ($a_1$–$a_3$) horizontal wind speed and ($b_1$–$b_3$) the vertical shear of horizontal wind speed, along with ($c_1$–$c_3$) the two-dimensional frequency distribution characteristics

of horizontal wind speed and the vertical shear of horizontal wind speed for H model (a$_1$, b$_1$, c$_1$), N-2D model (a$_2$, b$_2$, c$_2$) and D-H model (a$_3$, b$_3$, c$_3$) when the turbulent kinetic energy is negativie."

**For Fig. 3. This paper gives the result of taking four oblique beams as a total sample. Beacuse the results were relatively consistent, although the horizontal wind component of the north and south beams was concentrated in 0 to 20 m/s (Fig. R-3, Fig. R-4).**

"

[Figure]

Figure 5. Distribution of $R_a^-$ for the (**a$_1$**) H model, (**a$_2$**) N-2D model, and (a$_3$) D-H model in 2012. Panels (**b$_1$**)−(**b$_3$**) and (**c$_1$**)−(**c$_3$**) are the same as (a$_1$)–(a$_3$) but for the results of the three models in 2013 and 2014, respectively. The subgraph at the lower right corner of (**a$_1$, b$_1$, c$_1$**) is the same as (**a$_1$, b$_1$, c$_1$**), but for $log_{10}$ ($R_a^-$)."

[Figure]

Figure R-3. The same as Fig. 3, but for North beam.

[Figure]

190    Figure R-4. The same as Fig. 3, but for East beam.

**For Fig.4, the result is based on the three years of observational data from the east and west beams. Beacause the north-south component of horizontal wind speed in Beijing is distributed between 0 m s⁻¹ and 20 m s⁻¹.**

195  "

[Figure]

Figure 6. (a) Deviation profile of the data volume involved in the statistics and the mean value of the profile. The annual mean value is 34,130, the mean value in July is 3743, and the mean value in February is 3150. ($b_1$–$b_3$) Probability of N-TKE in each gate for the H model, N-2D model and D-H

200  model, respectively. Panels ($c_1, c_2$) and ($d_1, d_2$) are the median, upper and lower quartile profiles of horizontal wind speed and the vertical shear of horizontal wind speed, respectively. Black/red/blue represents the characteristics of the year/July/February, respectively. Three years of radar observational data from 2012 to 2014 were used in the statistics."

205 **For Fig. 5 and Table3, we take the observations of four oblique beams as a total sample.**

"

[Figure]

Figure 7. Profiles of (a) $\varepsilon$, (b) Kz, (c) B-V frequency, (d) observation spectrum width, (e) beam and shear broadening, and (f) spectrum width caused by turbulence. The solid line is the median and the
210 shaded area is the upper and lower quartiles. In panels (a, b, e, f), the black/red/blue solid lines and shaded areas represent the median and upper and lower quartiles of the H model/N-2D model/D-H model, respectively.

215

Table 3. Turbulence parameters of the Beijing MST radar (39.78°N, 116.95°E) at the range of 3–19.8 km.

| | H model, $c_1 = 0.45$ | N-2D model, $c_1 = 0.49$ | D-H model, $c_1 = 0.27$ |
|---|---|---|---|
| Median log ($\varepsilon$) (m$^2$ s$^{-3}$) | –3.2 (7 km) to –2.7(12 km) | –3.0 (7 km) to –2.6 (12 km) | –3.3 (7 km) to –2.8 (14 km) |
| Median log (Kz) (m$^2$ s$^{-1}$) | 0.3 to 0.7 | 0.4 to 0.7 | 0.1 to 0.5 |

220     ”

**For Fig.6, it is based on the east beam.**

"

[Figure]

225     Figure 6. N-TKE distribution of three models over the Beijing MST radar site in July 2014: (a) the

east-west component of horizontal wind; (b–d) area of N-TKE (green shading) for the east beam using the (b) H model, (c) N-2D model and (d) D-H model. The red scattered points are the tropopause."

230 **For Fig7, We take the datasets of east and west beams as one sample and that of north and south beams as the other sample.**

"

[Figure]

Figure 8. Distribution of $\varepsilon$ in the middle and low mode of the Beijing MST radar in the range of 3–
235 7.8 km from 2012–2014: $(a_1 - a_3)$ distributional characteristics of $\varepsilon$ in the H model, N-2D model and D-H model (mid mode); $(b_1 - b_3)$ as in $(a_1 - a_3)$ but for low-mode data. The gray bars are the result of a north and south beams, the yellow bar are based on the east and west beams.

"

 **RC2 referee comment 2**

**Specific comments**

**In the revised manuscript, the authors added a discussion on how to calculate the vertical shear of horizontal wind for estimating the shear broadening component, where the vertical shear is calculated as the vertical gradient of absolute value of horizontal wind vector. However, since**

245 **shear broadening comes from the variation of the radial velocity within the radar volume, only the beam direction component of the horizontal wind vector contributes to the broadening of the radar spectrum. In fact, Nastrom (1997) shows scatter plots of spectral widths of the eastward (northward) beam versus vertical shear of zonal (meridional) winds to explore the shear broadening effects of WSMR observations. I recommend the authors to recalculate shear**

250 **broadening effects.**

Response:

Thank you very much for your reminder and suggestions.

As you may have considered, we have considered the issue of "component of the horizontal wind" in the new version, and recalculated shear broadening effects.

255 There are two main jobs, 1. The preparations of calculating shear broadening effects. 2. The selection of statistical samples.

**1. Preparations**

This study used the the absolute value of the component of the horizontal wind vector, did

260 not overdiscuss the effect of wind direction, where $\frac{\partial u}{\partial z}$ contains positive and negative values.

**1.1 The beam direction component of the horizontal wind**

Only the beam direction component of the horizontal wind vector contributes to the broadening of the radar spectrum. So the correct value of wind shear should be $\frac{\partial u}{\partial z}_{\phi}$, where $\phi$ is the azimuth direction of the mean wind (Nastrom, 1997; Dehgan and Hocking, 2011). In this study, we take the zonal

265    (meridional) winds to explore the shear broadening effects of the east and west (north and south) beam.

The vertical shear of horizontal wind $\frac{\partial u}{\partial z}$ is as following:

For the east and west beams:

$$u = u_x, \frac{\partial u}{\partial z} = \frac{\partial u_x}{\partial z} \tag{6}$$

For the north and south beams:

270

$$u = v_y, \frac{\partial u}{\partial z} = \frac{\partial v_y}{\partial z} \tag{7}$$

where $u_x$ and $v_y$ is zonal and meridional wind, respectively.

**1.2 The directions of $u_x$ and $v_y$**

The models take the u as the horizontal wind speed(Nastrom, 1997; Dehgan and Hocking, 2011). In

275    fact, the direction of the component of the horizontal wind is indicated by a positive or negative number,

such as th positive value of $u_x$ means the wind blows from west to east. So we do some simulation

experiments for these three models, the results show that the directions of $u_x$ and $v_y$ have no effect

on the results of H model, and have very little effect on D-H model and N-2D model, as shown in

Fig.R-1.

280    **This study used the the absolute value of the component of the horizontal wind vector, did**

**not overdiscuss the effect of wind direction, where $\frac{\partial u}{\partial z}$ contains positive and negative values.**

**Simulation experiments:**

[Figure]

**Figure R-9** The $\sigma^2_{s\&b}$ estimated from the three models relate to the vertical shear and horizontal wind

speed. The horizontal wind speed is in the range of -80 to 80 m/s, per 4m/s. (a)(b) $\frac{\partial u}{\partial z}$ is in the range

of -0.02 to 0.02 s⁻¹, per 0.002 s⁻¹. (c)(d) $\frac{\partial u}{\partial z}$ has just 3 numbers, -0.01 s⁻¹, 0 s⁻¹, 0.01 s⁻¹. (a)(c) are $\sigma^2_{s\&b}$,

(b)(d) are log ($\sigma^2_{s\&b}$). (c)(d) red lines are $\frac{\partial u}{\partial z}$=-0.01 s⁻¹, blue lines are $\frac{\partial u}{\partial z}$=0 s⁻¹, purple lines are $\frac{\partial u}{\partial z}$=0.01

s⁻¹. (c) The stars are $\frac{\partial u}{\partial z} = 0$ s⁻¹. The broadening components estimated from H model is just related

to $|\frac{\partial u}{\partial z}|$, so it is the same line for H model taking $\frac{\partial u}{\partial z}$ as 0.01 s⁻¹ or -0.01 s⁻¹.

The modified content revised version(clean) for calculating shear broadening effects is as follows:
Line No:198‑208 (Section 2.3.1, Page No:6 of 22)

"In fact, only the beam direction component of the horizontal wind vector contributes to the

broadening of the radar spectrum. So the correct value of wind shear should be $\frac{\partial u}{\partial z}_\phi$, where $\phi$ is the

azimuth direction of the mean wind (Nastrom, 1997; Dehgan and Hocking, 2011). In this study, we

take the zonal (meridional) winds to explore the shear broadening effects of the east and west (north and south) beam. The vertical shear of horizontal wind $\frac{\partial u}{\partial z}$ is as following:

For the east and west beams:

$$u = u_x, \quad \frac{\partial u}{\partial z} = \frac{\partial u_x}{\partial z} \tag{6}$$

300    For the north and south beams:

$$u = v_y, \quad \frac{\partial u}{\partial z} = \frac{\partial v_y}{\partial z} \tag{7}$$

where $u_x$ and $v_y$ is zonal and meridional wind, respectively. And the directions of $u_x$ and $v_y$ have no effect on the results of H model, and have very little effect on D-H model and N-2D model. This study used the the absolute value of the component of the horizontal wind vector, did not overdiscuss

305    the effect of wind direction, where $\frac{\partial u}{\partial z}$ contains positive and negative values."

**2. The statistical samples – four oblique beams**

The "**Preparations**" showed clearly how to recalculated shear broadening effects. But how to choose the the statistical samples is another thing that needs careful consideration. We have the observational

310    datasets of four oblique beams, but there are obvious variation in the distributions of $u_x$ and $v_y$ (as shown in Fig. R-2). here $u_x$ and $v_y$ is zonal and meridional wind, respectively.

[Figure]

Figure R-10. Two-dimensional frequency distribution characteristics of horizontal wind speed and vertical shear of horizontal wind speed within the height range of 3–19.8km above the Beijing MST radar station from 2012 to 2014. (a)(b)(c) The east-west component of horizontal wind, (d)(e)(f) The north-south component of horizontal wind.

The east-west component of the horizontal wind speed over the radar site is distributed between 0 m s$^{-1}$ and 60 m s$^{-1}$, and the vertical shear of the horizontal wind speed ranges from −0.014 to 0.014 s$^{-1}$. The north-south component of the horizontal wind speed in Beijing is distributed between 0 m s$^{-1}$ and 20 m s$^{-1}$, and the vertical shear of the horizontal wind speed ranges from −0.014 to 0.014 s$^{-1}$.

As shown in Table R-1, for the east and west beams, the rates of N-TKE ($\sigma_t^2 < 0$) of the H model, N-2D model and D-H model are in the range of 27%–32%, 15%–21% and 9%–15%, respectively. And for the north and south beams, the rates are in the range of 5%–8%, 2%–4% and 0.6%–1.0%,. The probability that the turbulence spectrum width is less than 0 calculated by different oblique beams are different. But results of the symmetric beams (such as east and west beams/ north and south beams) are similar.

Table R-1. Total frequency of $\sigma_t^2 < 0$ in the range of 3–19.8 km.

| Beams | Time | Total numbers | H, $\sigma_t^2 < 0$ | N-2D, $\sigma_t^2 < 0$ | D-H, $\sigma_t^2 < 0$ |
|---|---|---|---|---|---|
| East | 2012 | 287490 | 78484 (27.30%) | 43253(15.05%) | 28067(9.76%) |
| | 2013 | 278317 | 76038(27.32%) | 43886(15.77%) | 27836(10.00%) |
| | 2014 | 311233 | 90633(29.12%) | 54988(17.67%) | 34219(10.99%) |
| West | 2012 | 288060 | 82821(28.75%) | 46925(16.29%) | 32467(11.27%) |
| | 2013 | 280769 | 82019(29.21%) | 48156(17.15%) | 32931(11.73%) |
| | 2014 | 313848 | 103226(32.89%) | 64997(20.71%) | 44683(14.24%) |
| North | 2012 | 102079 | 7924(7.76%) | 3870(3.79%) | 923(0.90%) |
| | 2013 | 84402 | 6377(7.56%) | 3206(3.81%) | 724(0.86%) |
| | 2014 | 92084 | 5900(6.41%) | 3115(3.38%) | 726(0.79%) |
| South | 2012 | 101288 | 6932(6.84%) | 3583(3.54%) | 985(0.97%) |
| | 2013 | 83418 | 5635(6.76%) | 2985(3.58%) | 696(0.83%) |
| | 2014 | 91535 | 5061(5.52%) | 2674(2.92%) | 573(0.63%) |

330

So if the distributions of $u_x$ and $v_y$ won't affect the statistical results, we take the observational datasets of four oblique beams as a total sample. Otherwise, we take the datasets of east and west beams as one sample and that of north and south beams as the other sample.

335

The modified content revised version(clean) for **The statistical samples** is as follows:

Line No:252 (Section 3.1, Page No:8 of 22)

**For Tabel 2. We take four samples for every year.**

340     "

Table 2. Total frequency of $\sigma_t^2 < 0$ in the range of 3–19.8 km.

| Beams | Time | Total numbers | H, $\sigma_t^2 < 0$ | N-2D, $\sigma_t^2 < 0$ | D-H, $\sigma_t^2 < 0$ |
|---|---|---|---|---|---|
| East | 2012 | 287490 | 78484 (27.30%) | 43253(15.05%) | 28067(9.76%) |
|  | 2013 | 278317 | 76038(27.32%) | 43886(15.77%) | 27836(10.00%) |
|  | 2014 | 311233 | 90633(29.12%) | 54988(17.67%) | 34219(10.99%) |
| West | 2012 | 288060 | 82821(28.75%) | 46925(16.29%) | 32467(11.27%) |
|  | 2013 | 280769 | 82019(29.21%) | 48156(17.15%) | 32931(11.73%) |
|  | 2014 | 313848 | 103226(32.89%) | 64997(20.71%) | 44683(14.24%) |
| North | 2012 | 102079 | 7924(7.76%) | 3870(3.79%) | 923(0.90%) |
|  | 2013 | 84402 | 6377(7.56%) | 3206(3.81%) | 724(0.86%) |
|  | 2014 | 92084 | 5900(6.41%) | 3115(3.38%) | 726(0.79%) |
| South | 2012 | 101288 | 6932(6.84%) | 3583(3.54%) | 985(0.97%) |
|  | 2013 | 83418 | 5635(6.76%) | 2985(3.58%) | 696(0.83%) |
|  | 2014 | 91535 | 5061(5.52%) | 2674(2.92%) | 573(0.63%) |

"

**For Fig1, We take the datasets of east and west beams as one sample and that of north and south beams as the other sample.**

"

[Figure]

350    Figure 11. Two-dimensional frequency distribution characteristics of horizontal wind speed and vertical shear of horizontal wind speed within the height range of 3–19.8km above the Beijing MST radar station from 2012 to 2014. (a)(b)(c) The east-west component of horizontal wind, (d)(e)(f) The north-south component of horizontal wind.

"

355

**For Fig2 This paper just gives the resultes of the east-west component of horizontal wind. Beacause the north-south component of horizontal wind speed in Beijing is distributed between 0 m s⁻¹ and 20 m s⁻¹.**

360
[Figure]

[Figure]

Figure 12. Frequency distribution of (a₁–a₃) horizontal wind speed and (b₁–b₃) the vertical shear of horizontal wind speed, along with (c₁–c₃) the two-dimensional frequency distribution characteristics

of horizontal wind speed and the vertical shear of horizontal wind speed for H model    (a₁, b₁, c₁), N-

2D model    (a₂, b₂, c₂) and D-H model (a₃, b₃, c₃) when the turbulent kinetic energy is negativie."

**For Fig. 3. This paper gives the result of taking four oblique beams as a total sample. Beacuse the results were relatively consistent, although the horizontal wind component of the north and south beams was concentrated in 0 to 20 m/s (Fig. R-3, Fig. R-4).**

"

[Figure]

Figure 13. Distribution of $R_a^-$ for the (**a₁**) H model, (**a₂**) N-2D model, and (a₃) D-H model in 2012. Panels (**b₁**)–(**b₃**) and (**c₁**)–(**c₃**) are the same as (a₁)–(a₃) but for the results of the three models in 2013 and 2014, respectively. The subgraph at the lower right corner of (**a₁**, **b₁**, **c₁**) is the same as (**a₁**, **b₁**, **c₁**), but for $log_{10}$ $(R_a^-)$."

[Figure]

375

Figure R-3. The same as Fig. 3, but for North beam.

[Figure]

Figure R-4. The same as Fig. 3, but for East beam.

380 **For Fig.4, the result is based on the three years of observational data from the east and west beams. Beacause the north-south component of horizontal wind speed in Beijing is distributed between 0 m s$^{-1}$ and 20 m s$^{-1}$.**

"

[Figure]

385 Figure 14. (a) Deviation profile of the data volume involved in the statistics and the mean value of the profile. The annual mean value is 34,130, the mean value in July is 3743, and the mean value in February is 3150. ($\mathbf{b_1} - \mathbf{b_3}$) Probability of N-TKE in each gate for the H model, N-2D model and D-H model, respectively. Panels ($\mathbf{c_1}, \mathbf{c_2}$) and ($\mathbf{d_1}, \mathbf{d_2}$) are the median, upper and lower quartile profiles of horizontal wind speed and the vertical shear of horizontal wind speed, respectively. Black/red/blue

390 represents the characteristics of the year/July/February, respectively. Three years of radar observational data from 2012 to 2014 were used in the statistics."

**For Fig. 5 and Table3, we take the observations of four oblique beams as a total sample.**

"

[Figure]

395

Figure 15. Profiles of (a) $\varepsilon$, (b) Kz, (c) B-V frequency, (d) observation spectrum width, (e) beam and

shear broadening, and (f) spectrum width caused by turbulence. The solid line is the median and the

shaded area is the upper and lower quartiles. In panels (a, b, e, f), the black/red/blue solid lines and

shaded areas represent the median and upper and lower quartiles of the H model/N-2D model/D-H

400      model, respectively.

405

Table 3. Turbulence parameters of the Beijing MST radar (39.78°N, 116.95°E) at the range of 3–19.8 km.

| | H model, $c_1 = 0.45$ | N-2D model, $c_1 = 0.49$ | D-H model, $c_1 = 0.27$ |
|---|---|---|---|
| Median log ($\varepsilon$) (m$^2$ s$^{-3}$) | –3.2 (7 km) to –2.7(12 km) | –3.0 (7 km) to –2.6 (12 km) | –3.3 (7 km) to –2.8 (14 km) |
| Median log (Kz) (m$^2$ s$^{-1}$) | 0.3 to 0.7 | 0.4 to 0.7 | 0.1 to 0.5 |

"

410  **For Fig.6, it is based on the east beam.**

"

[Figure]

Figure 6. N-TKE distribution of three models over the Beijing MST radar site in July 2014: (a) the

east-west component of horizontal wind; (b–d) area of N-TKE (green shading) for the east beam using the (b) H model, (c) N-2D model and (d) D-H model. The red scattered points are the tropopause."

**For Fig7, We take the datasets of east and west beams as one sample and that of north and south beams as the other sample.**

"

[Figure]

Figure 16. Distribution of $\varepsilon$ in the middle and low mode of the Beijing MST radar in the range of 3–7.8 km from 2012–2014: $(a_1 - a_3)$ distributional characteristics of $\varepsilon$ in the H model, N-2D model and D-H model (mid mode); $(b_1 - b_3)$ as in $(a_1 - a_3)$ but for low-mode data. The gray bars are the result of a north and south beams, the yellow bar are based on the east and west beams.

"